# Non-muscle myosins control radial glial basal endfeet to mediate interneuron organization

**Brooke R. D'Arcy**[1], **Ashley L. Lennox**[1], **Camila Manso Musso**[1], **Annalise Bracher**[1],
**Carla Escobar-Tomlienovich**[1], **Stephany Perez-Sanchez**[1], **Debra L. Silver**[1,2,3,4,5]*

1 Department of Molecular Genetics and Microbiology, Duke University Medical Center, Durham, North
Carolina, United States of America, 2 Department of Cell Biology, Duke University Medical Center, Durham,
North Carolina, United States of America, 3 Department of Neurobiology, Duke University Medical Center,
Durham, North Carolina, United States of America, 4 Duke Institute for Brain Sciences, Duke University
Medical Center, Durham, North Carolina, United States of America, 5 Duke Regeneration Center, Duke
University Medical Center, Durham, North Carolina, United States of America

* debra.silver@duke.edu

doi.org/10.1371/journal.pbio.3001926

Notre Dame, Center for Stem Cells and
Regenerative Medicine, UNITED STATES

**Data Availability Statement:** All relevant data are
within the paper and its Supporting Information
files.

## Abstract

Radial glial cells (RGCs) are essential for the generation and organization of neurons in the
cerebral cortex. RGCs have an elongated bipolar morphology with basal and apical endfeet
that reside in distinct niches. Yet, how this subcellular compartmentalization of RGCs con-
trols cortical development is largely unknown. Here, we employ in vivo proximity labeling, in
the mouse, using unfused BirA to generate the first subcellular proteome of RGCs and
uncover new principles governing local control of cortical development. We discover a
cohort of proteins that are significantly enriched in RGC basal endfeet, with MYH9 and
MYH10 among the most abundant. *Myh9* and *Myh10* transcripts also localize to endfeet
with distinct temporal dynamics. Although they each encode isoforms of non-muscle myosin
II heavy chain, *Myh9* and *Myh10* have drastically different requirements for RGC integrity.
*Myh9* loss from RGCs decreases branching complexity and causes endfoot protrusion
through the basement membrane. In contrast, *Myh10* controls endfoot adhesion, as
mutants have unattached apical and basal endfeet. Finally, we show that *Myh9-* and
*Myh10*-mediated regulation of RGC complexity and endfoot position non-cell autonomously
controls interneuron number and organization in the marginal zone. Our study demonstrates
the utility of in vivo proximity labeling for dissecting local control of complex systems and
reveals new mechanisms for dictating RGC integrity and cortical architecture.

## Introduction

Radial glial cells (RGCs) play a vital role in cortical development by regulating the production
and organization of diverse cell types, including neurons and glia [1,2]. These functions are
influenced by the RGCs' bipolar morphology. RGCs extend a long basal process to the pia; this
elongated structure can be several hundred micrometers long in mice and centimeters long in
humans. The basal process ends in subcellular compartments called basal endfeet that attach
to the basement membrane (BM) (**Fig 1A**). RGCs also maintain connections to the ventricle

**Funding:** This work is supported by R01NS110388, NIH to DLS; R01NS083897, NIH to DLS; R01NS120667, NIH to DLS; DST Spark grant, Duke to DLS; F31NS0933762, NIH to ALL and a NSF GRFP fellowship to BRD. The funders had no role in study design, data collection and analysis, decision to publish, or preparation of the manuscript.

**Competing interests:** The authors have declared that no competing interests exist.

**Abbreviations:** AUC, area under the curve; BM, basement membrane; bRG, basal radial glia; CFSE, carboxyfluorescein succinimidyl ester; CP, cortical plate; CR cells, Cajal–Retzius cells; DAVID, Database for Annotation, Visualization, and Integrated Discovery; DDA, data-dependent acquisition; ECM, extracellular matrix; GO, Gene Ontology; IUE, in utero electroporation; MF, molecular function; MZ, marginal zone; NM II, non-muscle myosin II; NMHC II, non-muscle myosin heavy chain II; oRG, outer radial glia; RGC, radial glial cell; smiFISH, single-molecule inexpensive fluorescent in situ hybridization; VZ, ventricular zone.

through apical endfeet with their cell body positioned in the ventricular zone (VZ). The radially oriented basal processes serve as scaffolds to guide newborn neurons as they migrate from their birth location in the VZ to the cortical plate (CP) [3]. As development progresses, RGCs become morphologically more complex, with increased endfoot number and branching at the distal end of the basal process [4,5]. Importantly, basal endfeet are conserved structures found not just in RGCs, but also in other essential neural progenitors of the cortex, including outer or basal radial glia (oRG/bRG) [6–10]. Thus, understanding these subcellular structures can give valuable insights into cortical development.

RGC basal endfeet have been broadly linked to several key events of brain development. They form a physical barrier at the pia, preventing over-migration of neurons [11,12]. Consistent with this, detachment of basal endfeet from the pia can cause neurodevelopmental disorders characterized by ectopic neurons such as Cobblestone Lissencephaly Type II [11–14]. Basal endfeet are embedded in a local niche composed of interneurons and Cajal–Retzius cells (CR cells) in the marginal zone (MZ) and blood vessels and fibroblasts (meninges) above the BM [15]. Extrinsic cues from this basal niche, including retinoic acid and fibroblast growth factors, can influence RGC proliferation [16,17]. Hence, from their physical position, basal endfeet may play a role in signaling and organizing surrounding cells of the niche [18–21]. The basal process and endfeet are also preferentially inherited by daughter progenitors, suggesting that they may influence cell fate [8,22,23]. However, how these basal structures are controlled and influence critical features of cortical development is largely unknown.

Emerging data suggest that gene expression within RGCs can be controlled at the subcellular level. A number of mRNAs have been shown to localize to basal endfeet including *Ccnd2*, and 115 transcripts bound by the RNA binding protein FMRP [24–27]. Previous work from our lab has established the active trafficking of mRNAs from the cell body to basal endfeet, where transcripts are competent to undergo local translation [27]. Despite this, few endfoot localized proteins have been identified and functionally interrogated. Dystroglycan is localized to basal endfeet at the protein level and is required for maintenance of the BM and prevention of neuronal over migration [12]. Analysis of another localized gene, *Arhgap11a*, revealed roles for locally produced ARHGAP11A in endfoot morphology [28]. These studies suggest the importance of subcellular control for cortical development. Yet, beyond just a few examples, the local proteome of RGC endfeet has never been characterized.

In this study, we leverage in vivo proximity labeling to determine the subcellular proteome of RGC basal endfeet. Using mouse genetics, we interrogate the subcellular function of 2 highly enriched protein coding isoforms of non-muscle myosin heavy chain II (NMHC II). We demonstrate that these isoforms have complementary and distinct requirements in maintaining the structural integrity of RGCs during development. Dysregulation of this morphology causes non-cell-autonomous alterations in the cellular composition and organization of the MZ. Collectively, our data illustrate a diverse repertoire of proteins that may influence cortical development via local functions. Furthermore, we show how cells of the nervous system utilize distinct protein isoforms to coordinate control of unique subcellular functions.

## Results

### BioID labels radial glial proteins in vivo in an unbiased fashion

Previously, a thorough biochemical characterization of radial glial endfeet had been intractable due to an inability to efficiently and specifically isolate protein from this small (<10 μm) subcellular compartment. We developed a method to circumvent this limitation by combining 2 techniques: endfoot microdissection [27] and proximity-labeling proteomics. Endfoot microdissection allows for mechanical separation of basal endfeet from the rest of the RGC by

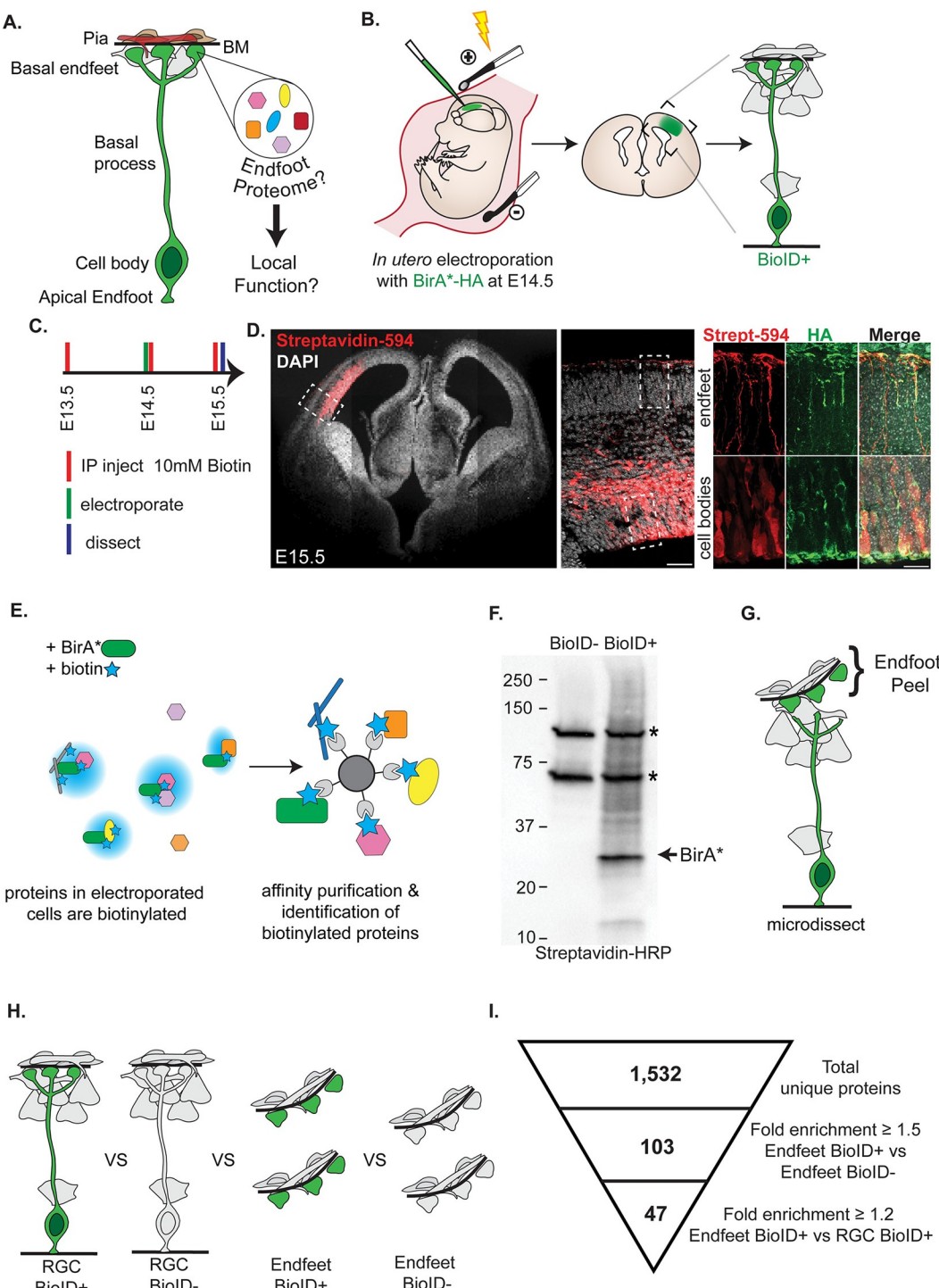

**Fig 1. In vivo proximity labeling reveals proteins enriched in radial glial basal endfeet compared to the whole cell.** (**A**) Cartoon representation of RGC morphology (green) and BM (black), and pial niche composed of interneurons and Cajal–Retzius cells (grey), blood vessels (red), and fibroblasts (brown). (**B**) Cartoon schematic of RGC labeling by IUE. (**C**) Experimental timing of IP biotin injections, electroporation, and dissection. (**D**) (Left) Staining with Streptavidin-594 (red) showing biotin labeling in electroporated cells of an E15.5 brain. (Right) Colocalization of biotin (red) with HA tagged BirA* protein (green) in the cell bodies and endfeet of RCGs. (**E**) Cartoon of (Left) BirA* (green) labeling all proteins with biotin (blue) and (Right) affinity purification of biotinylated proteins with streptavidin beads. (**F**) Western blot probed with Streptavidin-HRP showing biotinylated proteins including BirA*. Asterisks denote endogenously biotinylated carboxylases. (**G**) Cartoon showing microdissection to physically separate RGC endfeet from cell bodies. (**H**) The 4 samples compared in

our analysis (RGC BioID+, RGC BioID−, Endfeet BioID+, Endfeet BioID−). (**I**) Representation of criterion used to identify endfoot enriched proteins from the BioID data. $n$ = 3 biological replicates per condition, 66–68 pooled endfoot preparations per replicate for endfoot samples, 6 cortices pooled per replicate for whole RGC samples. Scale bars (**D**): 500 μm, 50 μm, 20 μm (left to right). BM, basement membrane; IP, intraperitoneal; IUE, in utero electroporation; RGC, radial glial cell.

peeling the outer layer, which includes BM, meninges, and basal endfeet, from the embryonic brain [27]. However, this approach alone cannot be used to distinguish endfoot-specific proteins from those of the extracellular matrix (ECM) and the meninges. Therefore, we turned to proximity labeling to tag proteins in RGCs. We chose to use BioID proximity-labeling due to its suitability for in vivo applications [29,30]. The BirA* enzyme releases reactive biotin intermediates that biotinylate lysine residues on neighboring proteins making it ideal for promiscuous biotinylation [31,32]. Canonically, BirA* can be fused to a protein-of-interest to characterize an interactome. However, we conceived a new use of this system. We predicted that, upon in utero electroporation (IUE) into the developing brain, unfused BirA* would freely diffuse through the cytoplasm to broadly label proteins within RGCs including in endfeet. This would allow us to classify RGC proteins that are enriched in BirA* (BioID+) cells compared to non-electroporated controls (BioID−) (**Fig 1B**).

We first verified that cytoplasmic BirA* specifically biotinylates proteins within electroporated RGCs including endfeet. We electroporated cytoplasmic BirA*-HA under control of the CAG promoter into embryonic day (E)14.5 cortices, where it would be preferentially expressed in progenitors of the VZ. Importantly, samples were harvested within 1 day of electroporation to minimize expression of BirA* in RGC progeny. Exogenous biotin was administered by consecutive intraperitoneal injections of 10 mM biotin into the pregnant dams at E13.5, E14.5 (at the time of electroporation), and E15.5 (shortly before dissection) (**Fig 1C**). Brains were harvested at E15.5, and immunofluorescence was used to detect the BirA* enzyme with anti-HA as well as biotinylated proteins with fluorescently conjugated streptavidin. The biotin signal was robust and restricted to the electroporated hemisphere, evident in RGC cell bodies and endfeet (**Fig 1D**). Similarly, HA-tagged BirA* also localized throughout the entire RGC. Importantly, the biotin signal was restricted to cells expressing the BirA* enzyme, demonstrating that biotinylation only took place in the presence of BirA* and ensuring the cellular specificity of this method.

Next, we verified that BirA* electroporated brains would enrich for biotinylated proteins, relative to endogenously biotinylated proteins from non-electroporated brains. Using the experimental approach described above, we collected samples from both BioID+ and BioID-brains for affinity purification (**Fig 1E**). In contrast to BioID− brains, the BioID+ brains contained many additional silver-stained bands (**S1A Fig**). We confirmed the biotinylation of these proteins by western blot with HRP-conjugated streptavidin (**Fig 1F**). Among these was a strong band at approximately 35 kDa corresponding to BirA*, which self biotinylates, as well as 2 proteins that correspond to endogenously biotinylated carboxylases [33,34]. These results demonstrate that electroporation of BirA* results in robust protein biotinylation that is readily distinguishable from that of the negative control.

To ensure that BirA* would label proteins with diverse subcellular localizations and molecular functions in an unbiased fashion, we performed qualitative tandem mass spectrometry on proteins affinity purified from BioID+ and BioID− cortices. In total, 1,453 unique proteins were identified, and of these, 961 were BioID+ specific. In the BioID+ sample, we detected proteins from diverse functional categories including cytoskeleton, RNA binding and proteostasis, vesicles and cell signaling, mitochondria and metabolism, and nuclear structures and DNA binding (**S1B Fig**). As expected, RGC-specific factors, including the intermediate filament

protein NES and the adhesion molecule NCAM1, were detected. This qualitative pilot experiment ensured that unfused BirA* would label functionally distinct classes of proteins across the cytoplasm.

## Discovery of an RGC basal endfoot enriched proteome

We next used quantitative proteomics to discover subcellularly localized proteins in RGCs. To purify biotinylated endfoot proteins, we used microdissection to remove the endfeet and surrounding meningeal tissue from electroporated brains (**Fig 1G**). We pooled these endfoot preparations from multiple cortices for each biological replicate for both electroporated (BioID+) and non-electroporated (BioID−) samples. To determine which proteins were specifically enriched in endfeet relative to the rest of the cytoplasm, we also collected non-microdissected (whole RGC) BioID+ and BioID− cortices (**Fig 1H**). For each condition (endfeet BioID +, endfeet BioID−, whole RGC BioID+, and whole RGC BioID−), 3 biological replicates were processed and affinity purified simultaneously before performing quantitative tandem mass spectrometry analysis. The quantitative proteomics analysis revealed 1,532 unique proteins across all samples (**S1–S3 Tables**). To identify proteins specific to RGCs, we filtered for proteins that were at least 1.5-fold enriched in the BioID+ condition relative to BioID− condition for both the endfoot and whole RGC samples (**Fig 1I**). This yielded 137 proteins from the endfoot samples and 245 from the whole RGCs (**S4 and S5 Tables**). To assess the reliability of these candidates, we evaluated several endfoot proteins by IF and mRNAs by smiFISH (**S2 Fig**). We validated PSME1 and PSEM2, which had $p$-values of 0.28 and 0.21, respectively (**S2G and S2H Fig**). Therefore, we further filtered the BioID+ candidates for those with a $p$-value $\leq 0.28$ (**S6 and S7 Tables**). This revealed 103 proteins present in endfeet but not necessarily subcellularly enriched (**S6 Table**). Next, we selected for proteins that were enriched in basal endfeet relative to the entire cytoplasm by filtering for those that were >1.2-fold higher in the BioID+ endfoot samples relative to the BioID+ whole RGC samples. As a result of this filtering, we discovered 47 proteins that are abundant and clearly enriched in endfeet relative to both the negative control and whole RGCs (**Fig 1I and Tables 1 and S8**).

Using STRING analysis, we investigated protein interaction networks of the endfoot-enriched proteins (**Fig 2A**) [35]. The endfoot proteome contained nodes related to actin and microtubule cytoskeletal regulation, ubiquitin pathway, mitochondria function, and RNA metabolism. It also contained a large interaction node of ECM proteins, including many collagens, laminins, and fibrinogens. Gene Ontology (GO) analysis reinforced these endfoot-enriched categories with ECM structural constituent, actin-dependent ATPase activity, and cell adhesion molecule binding among the most enriched molecular functions (**Fig 2B**). We validated several endfoot-enriched proteins, including TNS3 and ISG15, by co-immunofluorescence with EGFP- or mCherry-labelled RGC endfeet (**S2B–S2G Fig**). Notably, the mRNA encoding TNS3 was previously detected in the FMRP-bound transcriptome in endfeet [27]. Moreover, *If5, Map1b, and Kif21a*, also FMRP targets in endfeet, were detected in the endfoot proteome, although not specifically enriched compared to the whole RGC. We also validated additional endfoot-enriched proteins at the mRNA level by single-molecule inexpensive fluorescent in situ hybridization (smiFISH) including *Psme1, Col4a1*, and *Myl6* (**S2H–S2J Fig**). This suggests that at least a subset of endfoot-enriched proteins may be locally translated.

To interrogate the molecular similarity between endfeet and other subcellular structures in the nervous system, we compared our proteome to a previously published synapse proteome (**Fig 2C**) [36]. A total of 21 proteins were shared between the 2 datasets, accounting for 44.7% of the endfoot proteome including myosins, collagens, and proteasome subunits. Even more overlap was observed between our subcellular proteome and 2 independent synapse

**Table 1. Endfoot-enriched proteome uncovered by BioID.**

| Protein ID | ProteinTeller Probability | Fold Change (Endfeet+/RGC+) | Fold Change (Endfeet+/Endfeet−) | t test (Endfeet+/Endfeet−) | Protein Class |
| --- | --- | --- | --- | --- | --- |
| MYH9 | 1 | 43.67 | 3.49 | 0.085 | Myosin |
| CMPK2 | 0.91 | 34.91 | 13.07 | 0.009 | Mitochondria |
| KIAA1107 | 0.86 | 28.93 | 3.92 | 0.074 | Unknown |
| ISG15 | 1 | 24.94 | 37.80 | 0.000 | Ubiquitin |
| LAMC1 | 1 | 23.23 | 1.57 | 0.116 | ECM |
| MYH10 | 1 | 20.78 | 2.95 | 0.110 | Myosin |
| LCP1 | 1 | 18.68 | 4.04 | 0.004 | Actomyosin |
| FGA | 1 | 15.61 | 6.10 | 0.001 | ECM |
| CORO1A | 1 | 14.50 | 5.54 | 0.001 | Actomyosin |
| FN1 | 1 | 13.31 | 6.20 | 0.001 | ECM |
| MYL6 | 1 | 13.01 | 2.19 | 0.086 | Myosin |
| COL6A1 | 1 | 12.88 | 2.12 | 0.005 | ECM |
| POSTN | 1 | 12.43 | 4.41 | 0.034 | ECM |
| COL4A2 | 1 | 12.24 | 15.05 | 0.005 | ECM |
| TGM2 | 1 | 12.22 | 8.71 | 0.009 | Protein Crosslinking |
| H2-K1 | 0.97 | 12.05 | 18.71 | 0.001 | Signaling |
| HSPG2 | 1 | 10.84 | 3.25 | 0.000 | ECM |
| BST2 | 1 | 9.39 | 8.09 | 0.051 | Signaling |
| TGFBI | 1 | 8.90 | 14.10 | 0.013 | Signaling |
| TLN2 | 1 | 8.16 | 1.51 | 0.021 | Actin |
| FGG | 1 | 7.77 | 4.06 | 0.004 | ECM |
| MYH14 | 0.87 | 7.45 | 3.39 | 0.070 | Myosin |
| COL5A1 | 0.99 | 6.86 | 12.62 | 0.026 | ECM |
| COL4A1 | 0.86 | 6.40 | 9.20 | 0.004 | ECM |
| COL2A1 | 1 | 6.18 | 7.10 | 0.042 | ECM |
| COL1A1 | 1 | 6.09 | 7.69 | 0.017 | ECM |
| TNS3 | 0.99 | 5.78 | 5.32 | 0.000 | Actin Focal Adhesion |
| COL3A1 | 1 | 5.56 | 8.45 | 0.005 | ECM |
| COL1A2 | 1 | 5.39 | 8.82 | 0.009 | ECM |
| PSME2 | 1 | 5.36 | 2.17 | 0.214 | Ubiquitin |
| VPS29 | 0.97 | 4.11 | 1.66 | 0.174 | Vacuolar |
| LAMB2 | 0.99 | 3.90 | 1.51 | 0.045 | ECM |
| PSME1 | 1 | 3.48 | 1.67 | 0.283 | Ubiquitin |
| FBN2 | 0.99 | 3.42 | 7.31 | 0.011 | ECM |
| CNDP2 | 1 | 2.52 | 1.84 | 0.065 | Signaling Mapk |
| GGA1 | 1 | 2.47 | 2.67 | 0.159 | Ubiquitin |
| FGB | 1 | 2.13 | 1.64 | 0.053 | ECM |
| KIF15 | 0.91 | 1.89 | 2.58 | 0.016 | Microtubule |
| EDC4 | 1 | 1.62 | 2.06 | 0.013 | RBP |
| SAMHD1 | 1 | 1.48 | 2.10 | 0.005 | GTPase |
| RBBP9 | 0.87 | 1.46 | 2.76 | 0.013 | Hydrolase |
| RNF213 | 1 | 1.34 | 5.04 | 0.002 | Ubiquitin |
| CEP295 | 0.97 | 1.30 | 3.57 | 0.099 | Microtubule |
| ACAD9 | 1 | 1.27 | 1.71 | 0.241 | Mitochondria |
| PSA7L | 0.99 | 1.26 | 1.52 | 0.265 | Ubiquitin |
| DCTN1 | 1 | 1.24 | 4.27 | 0.000 | Microtubule |
| FERMT3 | 0.99 | N/A | 2.50 | 0.002 | Integrin |

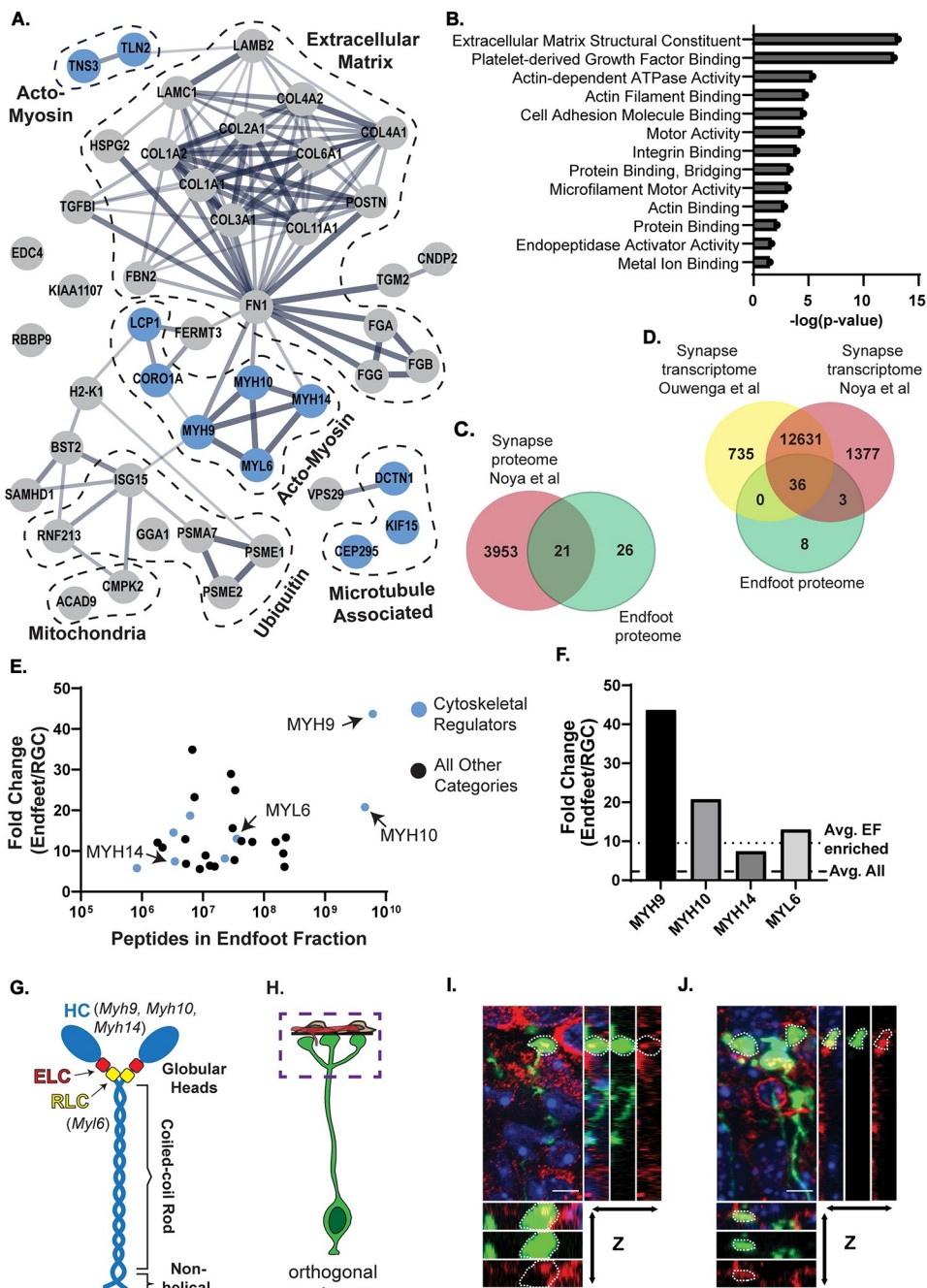

**Fig 2. The endoot proteome is enriched for ECM and cytoskeletal regulators, including NM IIA and B. (A)**
STRING analysis of the 47 endoot-enriched proteins. Dashed lines denote prominent protein classes. Cytoskeletal
regulators indicated as blue circles. (**B**) GO analysis of the endoot-enriched proteome showing all enriched categories
with $p \leq 0.05$. (**C**) Venn diagram comparing synapse proteome [36] (red) and endoot proteome (green). (**D**) Venn
diagram comparing synapse transcriptomes (yellow [37] and red [36]) and endoot proteome (green). (**E**) Scatter plot
depicting fold change vs. peptide count in the endoot fraction for 47 endoot-enriched proteins. Cytoskeletal
regulators indicated as blue circles. (**F**) Bar graph showing protein fold change of endfeet vs. RGC for non-muscle
myosin heavy and light chains, compared to average fold change of all proteins in endfeet (dashed line) and all endoot
enriched proteins (dotted line). (**G**) Cartoon representation of NM II structure, annotated with heavy and light chain
isoforms identified in endoot proteome. ELC, essential light chain, HC, heavy chain; RLC, regulatory light chain. (**H**)
Cartoon representation of RGC and pial niche with box outlining the location of images in **I** and **J**. (**I, J**) Colocalization
of MYH9 (red) (**I**) or MYH10 (red) (**J**) with endfeet (green) including orthogonal views. Endfeet marked by white
dotted lines. Scale bar (**I, J**): 5 μm. *n* = 3 brains, 3 sections per brain (**I, J**). Data underlying graphs included in S1 Data.
ECM, extracellular matrix; GO, Gene Ontology; NM II, non-muscle myosin II; RGC, radial glial cell.

transcriptomes (**Fig 2D**) [36,37]. This intersection suggests that there may be a cohort of conserved genes required for maintenance of distal, subcellular structures [38,39]. Furthermore, the non-overlapping genes may provide insights into RGC-specific components and functions.

Through these experiments, we have characterized the first subcellular proteome of RGC endfeet. We identified both intracellular and extracellular proteins, some of which are candidates for local translation in endfeet. These enriched proteins comprise functionally related categories that may provide further insights into subcellular structures and functions of RGCs.

## Non-muscle myosin II heavy chain isoforms are highly enriched in basal endfeet

NMHC II isoforms were among the most enriched proteins from the endfoot proteome (**Fig 2E and 2F**). Non-muscle myosin II (NM II) is a heteromeric complex composed of 2 heavy chains, 2 regulatory light chains, and 2 essential light chains (**Fig 2G**). The most highly enriched protein was NMHC IIA (MYH9), encoded by *Myh9*. MYH9 was enriched 43.7-fold in endfeet relative to the whole RGCs. Mice and humans have 2 additional heavy chain isoforms, NMHC IIB (MYH10) and NMHC IIC (MYH14), encoded by *Myh10* and *Myh14*, respectively. These isoforms were also highly enriched in endfeet with 20.8-fold enrichment for MYH10 and 7.5-fold enrichment MYH14 (**Fig 2E–2G**). These heavy chain components are highly conserved between mice and humans with 97%, 99%, and 91% amino acid homology for MYH9, MYH10, and MYH14, respectively. NM myosin II light chain 6 (MYL6) was also found in the endfoot proteome (**Table 1** and **Fig 2A, 2E and 2F**). Localization of MYH9, MYH10, and MYH14 to endfeet in E15.5 brains was validated by immunofluorescence (MYH9 and 10) and with a MYH14-GFP knock-in mouse [40] (**Figs 2I, 2J, S2A and S2B**).

In endfeet, MYH9 and MYH10 showed high protein levels and were some of the most enriched candidates, suggesting that they may be particularly important in RGCs. We focus our analysis on these isoforms because, in comparison, MYH14 had lower expression and less enrichment in endfeet (**Figs 2E, 2F, S2A, and S2B**). As part of the NM II hexamer, MYH9 and MYH10 are essential for actin cross-linking and contraction and implicated in cellular protrusions, cell migration, mitosis, and adhesion [41,42]. The enrichment of non-muscle myosins in basal endfeet suggests that they could locally control these cellular processes in RGCs. Previous studies have shown *Myh10* is essential for neural development, with *Myh10* null or mutant mice presenting with severe hydrocephalus, disrupted neuroepithelial adhesion, and disorganized neuronal migration [43–45]. While NMHC II proteins have been implicated in cortical development, their precise requirements in RGCs are unknown.

## *Myh9* and *Myh10* transcripts localize to basal endfeet with distinct temporal dynamics

The proteomic screen and subsequent validation indicate that endfeet are significantly enriched for NMHC II proteins, and show particularly high levels of MYH9 and MYH10. Previous data indicate that endfeet contain local transcriptomes and are capable of subcellular translation [24,27]. We therefore examined the subcellular localization of *Myh9* and *Myh10* transcripts using smiFISH within EGFP-labeled endfeet. At E15.5, both *Myh9* and *Myh10* were localized within endfeet (**Fig 3A and 3B**). To define the timing of this localization, we characterized the localization of *Myh9* and *Myh10* across different stages of cortical development, from E12.5 to E16.5. Both transcripts were expressed at all stages as evidenced by smiFISH puncta in the cell bodies (**Fig 3C, 3E, and 3G**). However, surprisingly, the subcellular distribution of *Myh9* and *Myh10* in RGC endfeet was distinct over the course of development.

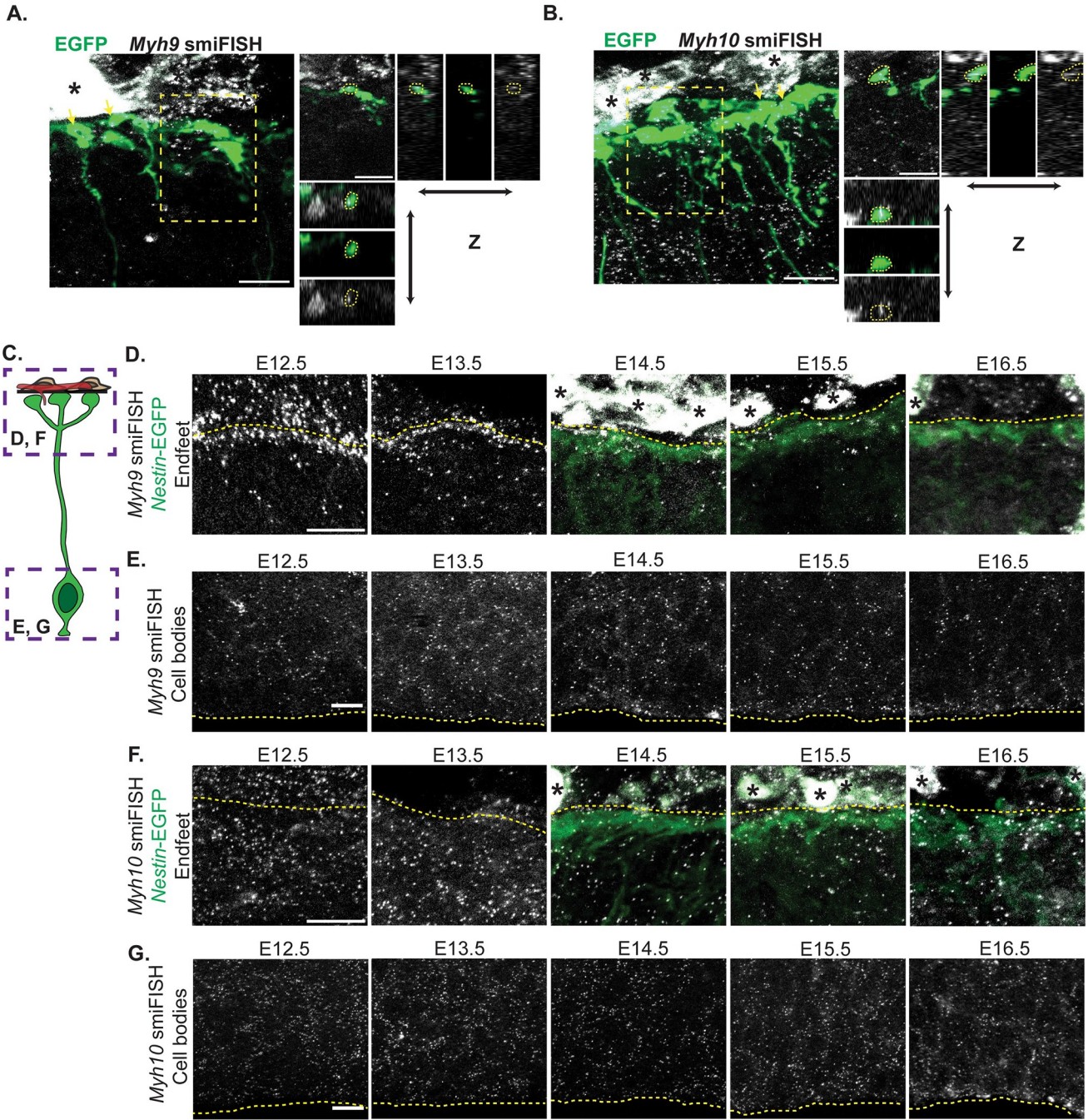

**Fig 3. *Myh9* and *Myh10* mRNAs localize to endfeet in complementary patterns that change across development. (A, B)** Colocalization of *Myh9* smiFISH (white, A) or *Myh10* smiFISH (white, B) with electroporated endfeet (green) at E15.5. (Left) Yellow arrows denote endfeet colocalized with smiFISH probes. Yellow box outlines area used for orthogonal views. (Right) Orthogonal views of colocalization. (**C**) Cartoon representation of RGC with boxes indicating cell body imaged in **E** and **G** and endfeet imaged in **D** and **F**. (**D**) Localization of *Myh9* smiFISH puncta (white) in endfeet from E12.5 to E16.5. E14.5 to E16.5 endfeet (green) labeled using *Nestin*-EGFP mouse [46]. Note, EGFP does not clearly label endfeet at E12.5 and E13.5. Yellow dotted lines denote the border between the endfeet (below) and the pia (above). (**E**) Localization of *Myh9* smiFISH puncta (white) in cell bodies from E12.5 to E16.5. Yellow dotted lines denote ventricular border. (**F**) Localization of *Myh10* smiFISH puncta (white) in endfeet from E12.5 to E16.5. E14.5 to E16.5 endfeet (green) labeled using *Nestin*-EGFP mouse. Yellow dotted lines denote the border between the endfeet (below) and the pia (above). (**G**) Localization of *Myh10* smiFISH puncta (white) in cell bodies from E12.5 to E16.5. Yellow dotted lines denote ventricular border. Samples observed for selection of representative images: *n* = 3 brains, 3 sections per brain (**A, B**). *n* = 2 to 4 brains, 3 sections per brain (**D, E, F, G**). Scale bars: 10 μm (**A, B, D, E, F, G**). Black asterisks denote background signal from the meninges (**A, B, D, F**). RGC, radial glial cell; smiFISH, single-molecule inexpensive fluorescent in situ hybridization.

*Myh9* mRNA was most enriched in endfeet at E12.5 and declined at E13.5 and E14.5, showing only sparse levels at E15.5 and E16.5 (**Fig 3C and 3D**). In contrast, endfoot localized *Myh10* showed relatively low localization at early developmental stages (E12.5) with gradual increases until the last stage examined (E16.5) (**Fig 3C and 3F**). We did not observe corresponding changes in *Myh9* and *Myh10* mRNA distribution in the cell body across development, suggesting that these changes are not driven solely by expression differences (**Fig 3E and 3G**). Our data suggest the intriguing possibility that *Myh9* and *10* could have unique and complementary roles in endfeet over the course of development. The observation that both mRNAs and proteins localize to endfeet supports the possibility that they may be locally produced and have local functions. It further proposes a role for isoform switching in endfeet, such that one isoform of NM II may be required in endfeet early in development and then may be replaced by the other key isoform.

## *Myh9* regulates RGC complexity and endfoot organization relative to the basement membrane

Due to the robust enrichment of MYH9 and presence of *Myh9* mRNA in endfeet, we hypothesized that it could be important for local RGC morphology and function. To test this, we crossed *Myh9*$^{lox/lox}$ and *Emx1*$^{Cre/+}$; *Myh9*$^{lox/+}$ mice to produce control (Ctrl) (*Myh9*$^{lox/lox}$; *Emx1*-Cre$^{+/+}$), cHet, and cKO littermates with *Myh9* specifically removed from RGCs and their progeny [47,48] (**S3A and S3B Fig**). First, we injected carboxyfluorescein succinimidyl ester (CFSE), a dye that covalently labels intracellular molecules [49], into the ventricles of E16.5 embryos (**Fig 4A**). The direct delivery of CFSE to the ventricle targeted its uptake to RGCs and allowed us to quickly label their morphology including the basal process and basal endfeet [50]. We then collected the brains 2 hours later. In the Ctrl and cHet brains, the basal processes were straight and radially oriented and the endfeet formed a single, uninterrupted line at the pia (**Fig 4B**). In contrast, there were obvious gaps between individual *Myh9* cKO endfeet with many extended past their normal stopping point and into the pia (**Fig 4B**). These results indicate that *Myh9* mutants have defects in RGC endfeet.

To determine whether these morphology defects could be secondary to cell body defects from *Myh9* loss, we assessed neurogenesis. At the same stage, E16.5, there was no significant difference in the density of RGCs (SOX2+) nor in the number of apoptotic cells (CC3+) (**Figs 4C, 4F, and S3C**). Additionally, the organization of progenitors was normal, as shown by equivalent measurements of the SOX2+ cell layer thickness (**Fig 4E**). Furthermore, cortical thickness was similar across genotypes, also supporting the lack of a neurogenesis defect (**Fig 4D**). This indicates that *Myh9* is not required in RGCs for their number, survival, and organization at E16.5, suggesting that the impaired morphology is not due to cell body defects.

By E16.5, RGCs have complex basal structures including extensive branching and multiple endfeet [4,28]. To examine the requirement of *Myh9* for basal morphology at higher resolution, RGCs were sparsely labelled using IUE of membrane-bound EGFP (pGLAST-EGFP CAAX) at E15.5. We examined E16.5 brains and made 3D reconstructions of individual cells, focusing on branching in the MZ (**Fig 4G**). As an endfoot phenotype was not observed in the CFSE-labeled cHet brains (**Fig 4B**), we focused our in-depth analysis on the cKOs. First, we quantified the total number of branches for each RGC and found that the cKO RGCs have significantly fewer branches compared to their Ctrl counterparts (**Fig 4H and 4I**). Next, we assessed the complexity of the branches present by assigning first, second, third, fourth, fifth, or sixth branch order to each based on the number of ramifications between the basal process and the branch of interest (**Fig 4H and 4J**). The branches of *Myh9* cKO RGCs were less complex than those of Ctrl with significantly fewer second and third order branches (**Fig 4J**).

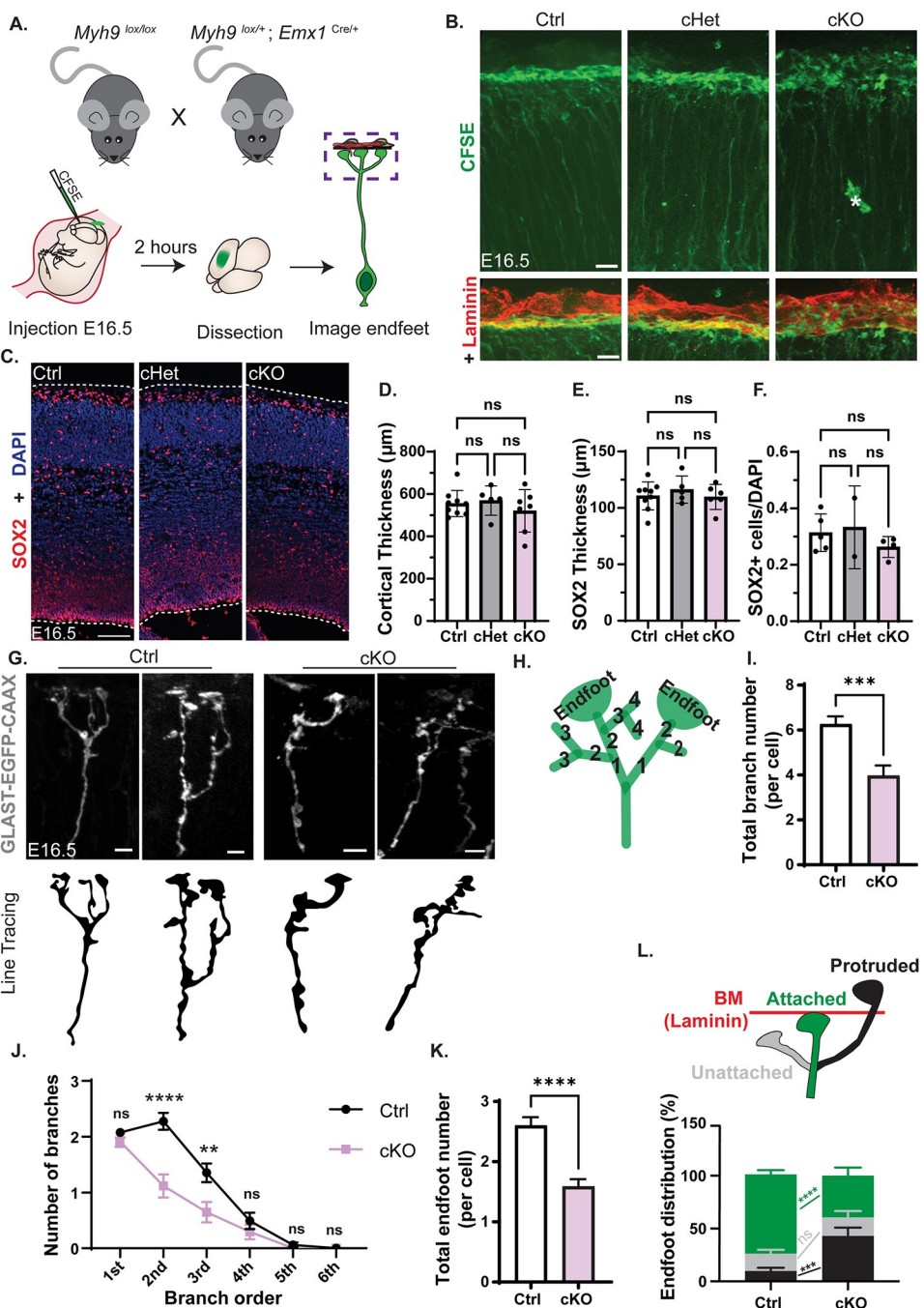

**Fig 4. *Myh9* regulates basal endfoot complexity and organization relative to the BM.** (**A**) Cartoon schematic of (Top) mating scheme used to generate embryos and (Bottom) method for CFSE labeling of RGC morphology. (**B**) CFSE labeling of E16.5 RGCs including endfeet (green) for *Myh9* Ctrl, cHet, and cKO brains (Top), with BM marked by laminin (red) (Bottom). Asterisk denotes background signal from vasculature. *n* = 5 Ctrl, 5 cHet, 4 cKO from 3 litters. (**C**) Cortical columns of E16.5 *Myh9* Ctrl, cHet, cKO brains stained for SOX2 (red) and DAPI (blue). White dashed lines denote tissue borders. *n* = 9 Ctrl, 5 cHet, 6 cKO from 3 litters. (**D**) Quantification of cortical thickness. *n* = 9 Ctrl, 5 cHet, 7 cKO from 3 litters. (**E**) Quantification of the thickness of the SOX+ layer. *n* = 9 Ctrl, 5 cHet, 6 cKO. (**F**) Quantification of SOX2+ cell density as the fraction of total DAPI cells. *n* = 5 Ctrl, 2 cHet, 4 cKO. (**G**) Representative images of 3D reconstructed Ctrl and cKO RGCs labeled with GLAST-EGFP-CAAX by IUE. (Top) Fluorescent images of EGFP+ basal processes and endfeet. (Bottom) Line tracing of top images. (**H**) Schematic detailing endfoot number and branch order quantification criteria for panels **I-K**. (**I**) Quantification of total branch number per cell. (**J**) Quantification of number of branches per branch order. (**K**) Quantification of total endfoot number per cell. (**L**) (Top) Schematic detailing quantification criteria of endfoot position. Endfeet were classified as

unattached (grey), attached (green), or protruded (black) relative to the BM (red) stained with laminin. (Bottom) Quantification of endfoot position represented as % of total endfeet analyzed. $n$ = 65 cells from 5 brains from 3 litters for Ctrl and $n$ = 34 cells from 3 brains from 2 litters for cKO (G-L) Error bars: SD (**D**, **E**, **F**), SEM. (**I**, **J**, **K**, **L**). One-way ANOVA with Tukey's (**D**, **E**, **F**), Student unpaired, two-tailed $t$ test (**I**, **J**, **K**) two-way ANOVA with Sidak's multiple comparison test (**L**) ns $p > 0.05$, *$p \leq 0.05$, **$p \leq 0.01$, ***$p \leq 0.001$, ****$p \leq 0.0001$. Scale bars: (**B**) 10 μm; (**C**) 100 μm; (**G**) 5 μm. Data underlying graphs included in S2 Data. BM, basement membrane; CFSE, carboxyfluorescein succinimidyl ester; IUE, in utero electroporation; RGC, radial glial cell.

Additionally, *Myh9* cKO RGCs had fewer endfeet per cell (**Fig 4K**). Altogether, these data reveal an essential requirement of *Myh9* for RGC complexity and endfoot number.

Next, we assessed the organization of RGC basal endfeet relative to the BM. For this, we categorized each endfoot as attached (connected to the BM), unattached (below the BM, but not touching it), or protruded (through the BM into the pia) (**Figs 4L and S3D**). Ctrl and cKO RGCs showed similar percentages of unattached endfeet, with 17% and 15%, respectively (**Fig 4L**). However, there was a significant striking difference in endfoot attachment, with 76% of Ctrl endfeet attached and only 42% of cKO. Compared to 43% of cKO endfeet, 8% of Ctrl endfeet protruded. This suggests that endfeet that fail to attach properly are instead protruding through the BM. Taken together with the CFSE labeling (**Fig 4B**), these data indicate that *Myh9* is required for proper endfoot organization and morphology. Considering the 43-fold enrichment of MYH9 protein (**Fig 2E and 2F**), localized mRNA (**Fig 3A and 3D**), and the lack of cell body defects (**Fig 4C–4F**), our results are consistent with a subcellular role for MYH9 in RGC complexity and endfoot organization.

## *Myh10* is required to maintain RGC basal and apical endfeet attachments

In addition to MYH9 (NMHC IIA), MYH10 (NMHC IIB) is also significantly and abundantly enriched in endfeet. We therefore sought to investigate the specificity and necessity of this additional NMHC II isoform, MYH10, in endfeet. Following conditional removal of *Myh10* from RGCs and their progeny with *Emx1*-Cre (**S4A Fig**), we labeled RGCs with in utero injections of CFSE to visualize morphology [48,51] (**Fig 5A**). At E16.5, we observed a profound loss of endfeet at the BM in the *Myh10* cKO cortices, compared to Ctrl (*Myh10*$^{lox/lox}$; *Emx1*-Cre$^{+/+}$) or cHets (**Fig 5B**). In these brains, the basal processes lost their radially oriented organization (**Fig 5C**). We then assessed earlier developmental stages to understand if endfeet are initially attached to the pia and then detach or fail to form properly. At E14.5, many endfeet were organized at the pia, compared to E16.5 (**Fig 5D**). By E15.5, progressively more endfeet became unattached from the BM (**Fig 5E**). This indicates that while *Myh10* is not initially required for endfeet attachment at the BM, it is necessary to maintain their adhesion as development progresses. Notably, this phenotype was drastically different than that of *Myh9* cKO brains.

As *Emx1*-Cre depletes *Myh10* from RGC cell bodies as well as endfeet, we interrogated the requirement of *Myh10* for progenitor survival and neurogenesis. At E16.5, there was no significant difference in the density of SOX2+ RGCs in the cortex (**Fig 5F and 5H**). Widespread cell death was also not observed (**S4B Fig**). Additionally, cortical thickness was similar between Ctrl, cHet, and cKO brains (**Fig 5G**). Altogether, this suggests that, similar to *Myh9*, *Myh10* is not essential for neurogenesis at E16.5. Furthermore, it indicates that the reduction of basal endfeet at the BM is not a result of RGC cell death or decreased RGC number.

While the number of SOX2+ progenitors was unaffected, their organization was disrupted with fewer cells in the VZ and more located basally throughout the CP (**Fig 5F**). The proliferative capacity of these disorganized progenitors was maintained as shown by the distribution of Ki67+ cells throughout the cortex (**S4C Fig**). To understand the cause of this mispositioning, we examined earlier stages of development. The SOX2+ RGCs were initially properly

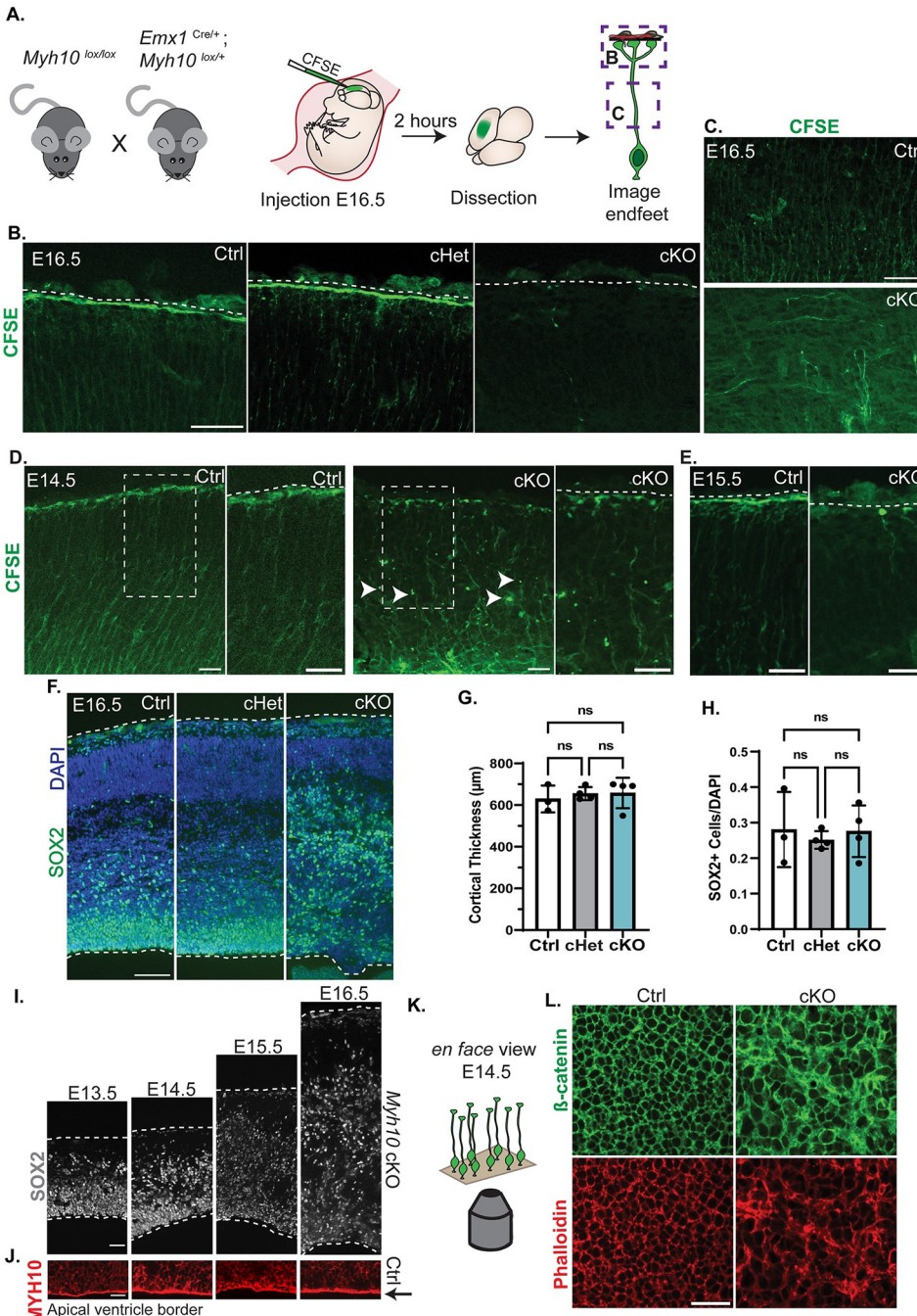

**Fig 5. *Myh10* is required for attachment of both basal and apical endfeet.** (**A**) Cartoon schematic of (Left) mating scheme used to generate embryos, (Center) method for CFSE labeling of RGC morphology, and (Right) purple boxes representing regions in **B** and **C**. (**B**) CFSE labeling of E16.5 RGCs (green) for *Myh10* Ctrl, cHet, and cKO brains. White dotted line denotes border between endfeet and pia. *n* = 3 Ctrl, 3 cHet, 4 cKO from 2 litters. (**C**) Comparison of basal process morphology between E16.5 *Myh10* Ctrl and cKO RGCs labeled with CFSE. *n* = 3 per genotype from 1 litter. (**D**) Endfoot position and attachment in *Myh10* cKO and Ctrl E14.5 brains labeled with CFSE (green). Arrow heads denote detached endfeet. Dotted box marks region of interest for right panel of each genotype. White dotted line denotes border between endfeet and pia. *n* = 4 per genotype from 1 litter. (**E**) Ctrl and *Myh10* cKO endfeet labeled with CFSE (green) at E15.5. White dotted line denotes border between endfeet and pia. *n* = 3 per genotype from 2 litters. (**F**) Cortical columns of E16.5 *Myh10* Ctrl, cHet, and cKO brains, stained with SOX2 (green) and DAPI (blue). White dotted lines mark tissue borders. *n* = 3 Ctrl, 4 cHet, and 4 cKO from 2 litters. (**G**) Quantification of cortical thickness. (**H**) Quantification of SOX2+ cell density calculated as the fraction of total DAPI cells. (**I**) SOX2+ (grey) cell position across development from E13.5 to E16.5 in *Myh10* cKO cortical columns. White dotted lines denote pial and

ventricular boundaries. $n$ = 3 E13.5, 3 E14.5, 4 E15.5, 6 E16.5. From 1 litter for E13.5 to E15.5 and 2 litters for E16.5. (**J**) MYH10 staining (red) of E13.5 to E16.5 Ctrl brains with imaging focused on apical endfeet at the ventricle. Arrow indicates MYH10+ staining in apical endfeet at the ventricular border of coronal sections. $n$ = 3 per stage. (**K**) Cartoon depiction of en face imaging of apical endfeet at E14.5 shown in **L**. (**L**) En face imaging of apical endfeet in E14.5 *Myh10* Ctrl and cKO cortices, labeled with anti β-catenin (green) and Phalloidin (red) to mark actin. $n$ = 9 Ctrl and 7 cKO from 3 litters. Error bars: SD. (**G, H**) One-way ANOVA with Tukey's (**G, H**) ns $p$ > 0.05. Scale bar: (**B, C, I**) 50 μm; (**D, E**), 10 μm; (**F**) 100 μm; (**J, L**) 25 μm. Data underlying graphs included in S3 Data. CFSE, carboxyfluorescein succinimidyl ester; RGC, radial glial cell.

organized at E13.5; however, as development progressed, they became increasingly disorganized and displaced from the ventricle (**Fig 5I**).

RGCs are anchored to the ventricle by an apical endfoot (**Fig 1A**) [52,53]. Indeed, basal displacement of progenitors is also seen in mutants where apical endfeet are impaired [54,55]. Therefore, we hypothesized that MYH10 may influence apical endfoot attachment to mediate RGC localization in the VZ. To test this, we first examined MYH10 expression at the ventricle. This revealed that, in addition to basal endfoot localization, MYH10 is also strongly localized at the apical border across development (E13.5 to E16.5) (**Fig 5J**). This apical signal was abolished in *Myh10* cKO brains (**S4A Fig**). We then used en face imaging of the cortices to assess apical endfoot morphology with phalloidin to mark F-actin and β-catenin to mark cell–cell adhesions (**Fig 5K**). We did this analysis at E14.5, a stage when some RGCs were still in the VZ and others were basally positioned (**Fig 5I**). Both F-actin and β-catenin showed clear disorganization of the apical endfeet in the cKO brains, while the cHets were indistinguishable from Ctrl (**Figs 5L and S4D**). These results suggest that MYH10 localization at apical endfeet enables RGCs to remain attached to the ventricle and indicates that loss of *Myh10* impairs actin organization. Our data suggest that this role in the apical endfoot is unique to *Myh10*; unlike MYH10, MYH9 is not enriched in the apical endfoot and *Myh9* cKO RGCs are not basally displaced.

The gradual loss of basal endfeet from the BM and apical endfeet from the ventricular border demonstrates that MYH10 is essential for the attachment of both basal and apical endfeet. The divergent phenotypes of *Myh9* and *Myh10* cKO RGCs highlight the unique functions of each NMMHC II isoform in endfeet and cortical development.

## RGC integrity impacts interneuron number and organization in the MZ

Our data show that MYH10 is essential for RGC basal endfeet attachment to the BM. We thus sought to understand whether this endfoot phenotype impacted surrounding cells, with a goal of gaining new insights into functions of RGC endfeet. In E16.5 *Myh10* cKO brains, almost all basal endfeet were detached from the BM and absent from their niche in the MZ [56]. We thus examined the general architecture of the MZ that is populated by CR cells and interneurons (**Fig 6A**). First, we used DAPI staining to label all nuclei (**Fig 6B**). We discovered 33% more DAPI+ cells in the MZ of *Myh10* cKO brains compared to Ctrl (**Fig 6C**). However, the thickness of the MZ was unchanged, indicating the influx of additional cells caused an overall higher cell density (**Fig 6D**). To assess the identity of these cells, we stained brains for calretinin, which labels both CR cells and interneurons at this stage [57–59]. We quantified a 2.4-fold increase in the thickness of the calretinin layer in the *Myh10* cKO brains, suggesting that cells within this region may be disorganized and/or increased in number (**Fig 6E and 6F**). To determine if this defect preceded the endfoot phenotype, we assessed both Calretinin and Reelin at E13.5. Neither were impaired at this earlier stage, supporting the idea that endfeet defects underlie altered MZ composition (**S5B and S5C Fig**). To further identify the fate of these cells, we quantified P73, a marker for many CR cell subtypes [60–64]. Notably P73+ cells were not

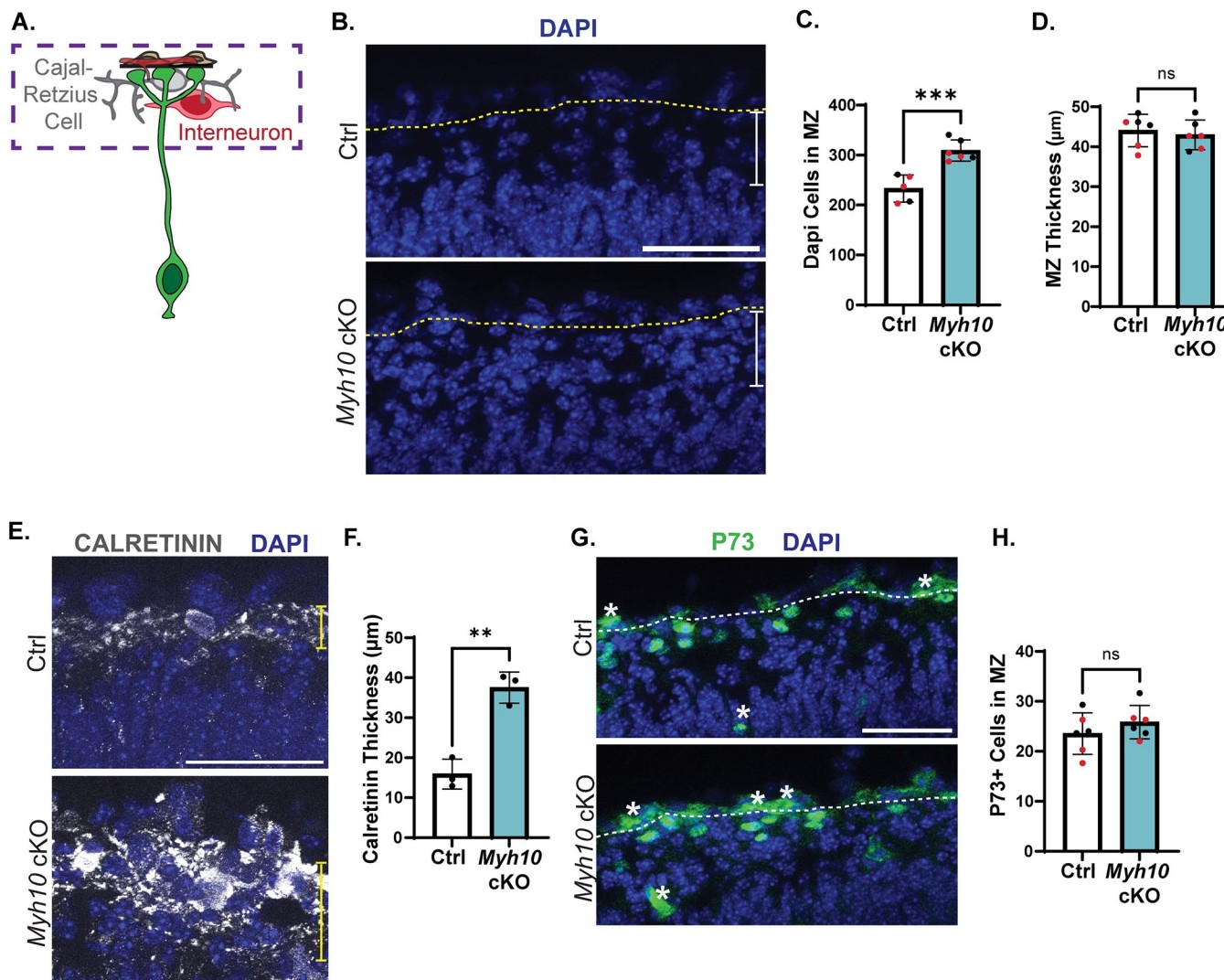

**Fig 6. Detached endfeet, caused by loss of *Myh10*, impact cortical architecture.** (**A**) Cartoon schematic of an RGC (green) and the local niche where the endfeet reside made up of BM (black), fibroblasts (brown), vasculature (red), CR cells (grey), and interneurons (red). Box indicates MZ region of images in **B**, **E**, **G**. (**B**) Labeling of all nuclei with DAPI (blue) in E16.5 *Myh10* Ctrl and cKO brains. Boundary between pia and MZ marked by yellow dashed line. White bar represents MZ thickness measurements in **D**. (**C**) Quantification of DAPI cells in the MZ. $n = 5$ Ctrl, 6 cKO from 2 litters. (**D**) Quantification of MZ thickness. Each data point represents the average of 3 measurements per section and 3 sections per brain. $n = 6$ per genotype from 2 litters. (**E**) Staining with Calretinin (white) to label CR cells and interneurons and DAPI (blue) to mark all nuclei in E16.5 *Myh10* Ctrl and cKO brains. Yellow bar represents where Calretinin thickness was measured for **F**. (**F**) Quantification of Calretinin thickness. $n = 3$ per genotype from 1 litter (**G**) P73 staining (green) of CR cell nuclei and DAPI (blue). Asterisks denote vasculature. White dotted line marks boundary between pia and MZ. (**H**) Quantification of P73+ cells in MZ. $n = 6$ per genotype from 2 litters Error bars: SD. (**C**, **D**, **F**, **H**). Student unpaired, two-tailed *t* test (**C**, **D**, **F**, **H**). Data points color-coded by litter (**C**, **D**, **F**, **H**). ns $p > 0.05$, *$p \leq 0.05$, **$p \leq 0.01$, ***$p \leq 0.001$, ****$p \leq 0.0001$. Scale bar: (**B**, **E**, **G**), 50 μm. Data underlying graphs included in S4 Data. BM, basement membrane; CR cells, Cajal–Retzius cells; MZ, marginal zone; RGC, radial glial cell.

increased in the MZ of *Myh10* cKO brains (**Fig 6G and 6H**). Likewise, there was also no difference in the thickness of the Reelin layer, which is expressed by CR cells at this stage [65] (**S5A Fig**). Together with the P73 data, this suggests that CR cell number is intact in the absence of endfeet in *Myh10* cKO brains. We then stained for interneurons using LHX6 (**Fig 7A**). This revealed the identity of the DAPI+ surplus cells to be interneurons with 43% more LHX6 + cells in the MZ of *Myh10* cKO brains (**Fig 7A and 7B**). As *Emx1*-Cre is not active in these interneurons, this indicates that *Myh10* controls interneuron organization in the MZ by acting

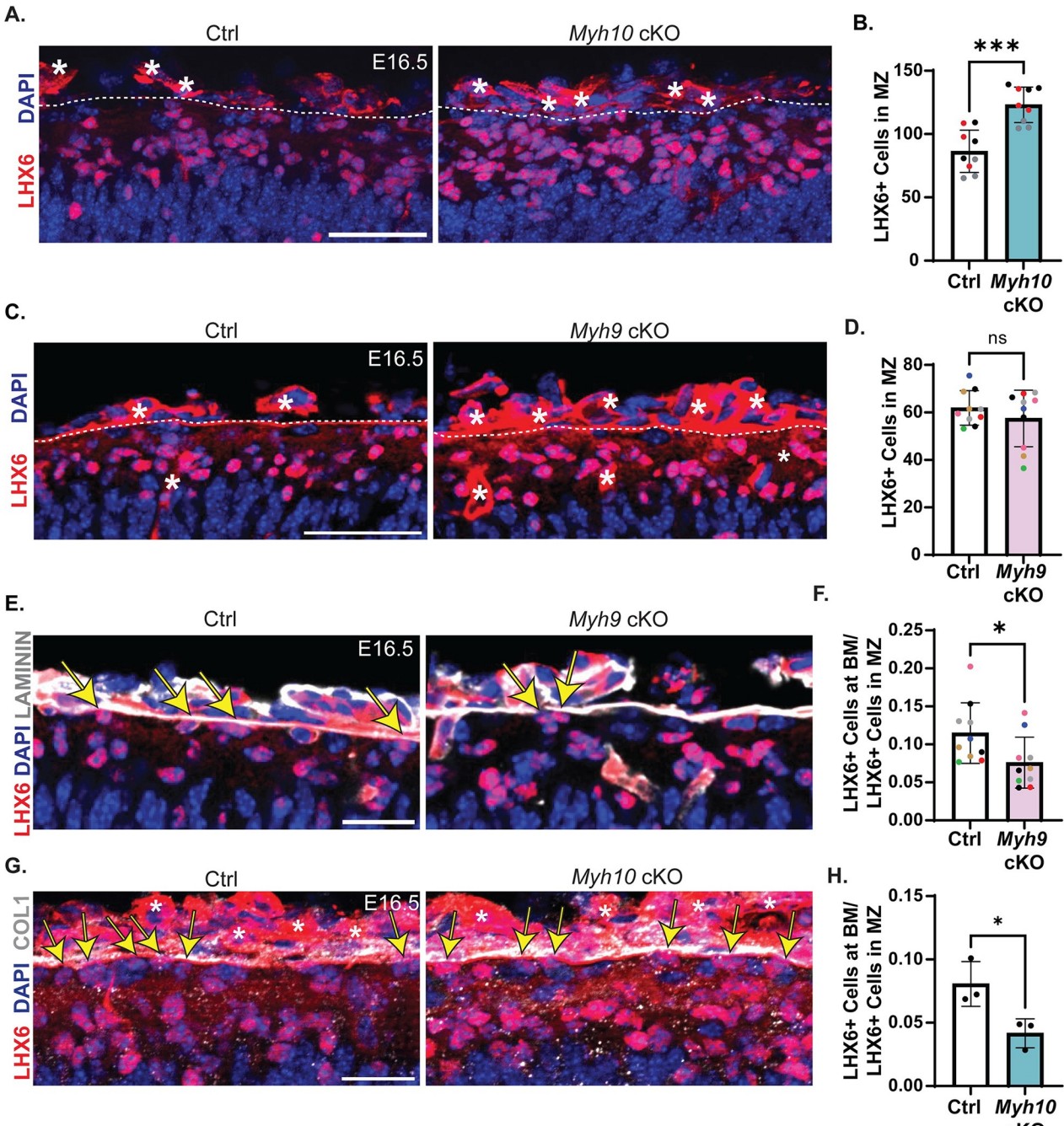

**Fig 7. RGC morphology, regulated by *Myh9* and *Myh10*, influences interneuron number and organization in the MZ.** (**A**) Staining with LHX6 (red) to label interneurons and DAPI (blue) to mark all nuclei. *Myh10* Ctrl and cKO brains at E16.5. Asterisks denote vasculature. White dotted line marks boundary between pia and MZ. (**B**) Quantification of LHX6+ cells in the MZ. *n* = 9 per genotype from 4 litters. (**C**) Staining with LHX6+ (red) and DAPI (blue) in the MZ of Ctrl and *Myh9* cKO E16.5 brains. (**D**) Quantification of the number of LHX6+ cells in the MZ of Ctrl and *Myh9* cKO brains. *n* = 10 per genotype from 6 litters. (**E**) Comparison of LHX6+ (red) cells touching the BM in Ctrl and *Myh9* cKO brains. BM labeled with laminin (white) and all nuclei with DAPI (blue). Yellow arrows indicate LHX6+ cells touching the BM. (**F**) Quantification of the fraction of LHX6 + cells in the MZ touching the BM compared to total LHX6+ cells in the MZ. *n* = 10 per genotype from 6 litters. (**G**) Comparison of LHX6+ (red) cells touching the BM in Ctrl and *Myh10* cKO brains. BM labeled with COL1 (white) and all nuclei with DAPI (blue). Yellow arrows indicated LHX6 + cells touching the BM. (**H**) Quantification of the fraction of LHX6+ cells in the MZ touching the BM compared to total LHX6+ cells in the MZ. *n* = 3 per genotype from 1 litter. Error bars: SD. (**B, D, F, H**). Student unpaired, two-tailed *t* test (**B, D, F, H**). Data points color-coded by litter (**B, D, F, H**). ns $p > 0.05$, *$p \leq 0.05$, **$p \leq 0.01$, ***$p \leq 0.001$, ****$p \leq 0.0001$. Scale bar: (**A, C**) 50 μm; (**E, G**) 25 μm. Data underlying graphs included in S5 Data. BM, basement membrane; COL1, Collagen 1; MZ, marginal zone; RGC, radial glial cell.

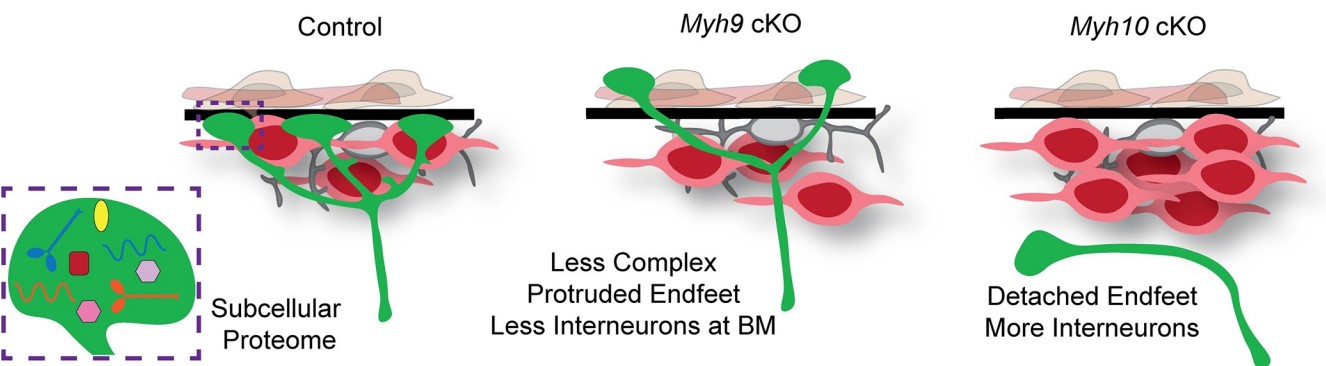

**Fig 8. Endfoot-enriched proteome highlights MYH9 and MYH10, which control RGC integrity and interneurons.** Cartoon summary illustrating the local proteome of RGC basal endfeet. The middle and right cartoons depict the impact of *Myh9* and *Myh10* on RGC (green) morphology and the organization of CR cells (grey) and interneurons (red). BM, basement membrane; CR cells, Cajal–Retzius cells; RGC, radial glial cell.

non-cell autonomously in RGCs. These data further show that RGC endfoot attachment is essential for maintaining proper number of interneurons in the developing cortex.

Our data show that MYH9 controls unique aspects of RGC morphology and endfoot position, compared to MYH10, which promotes RGC endfoot adhesion to the BM. Given these complementary phenotypes, this presented an opportunity to discover how endfeet influence interneurons. *Myh10* cKO brains had approximately 40% more LHX6+ cells in the MZ relative to Ctrl (**Fig 7A and 7B**). In contrast, there was no significant difference in the total number of LHX6+ cells in the MZ between Ctrl and *Myh9* cKO brains (**Fig 7C and 7D**). This suggests that in the absence of endfeet (*Myh10* cKO), more interneurons flood into the open space of the MZ. In comparison, reduced complexity of RGCs (*Myh9* cKO) does not affect interneuron number in the MZ.

In addition to their high density in the MZ, some interneurons are in tight contact with the BM. *Myh9*-deficient RGC endfeet have altered position and complexity relative to the BM. We therefore tested the possibility that these defects would instead influence interneuron position. To measure this, we calculated the fraction of LHX6+ cells touching the BM relative to all LHX6+ cells in the MZ. In Ctrl brains, a subset of LHX6+ interneurons in the MZ directly contact the BM (marked by laminin) [66]. In contrast, in *Myh9* cKO brains, there was a significant reduction in the fraction of LHX6+ cells touching the BM (**Fig 7E and 7F**). Similarly, the *Myh10* cKO mice, where endfeet were completely missing from the MZ, showed a significant reduction in the fraction of LHX6+ cells contacting the BM (**Fig 7G and 7H**). Taken together, these data demonstrate that endfeet are essential for both interneuron number and organization in the MZ. This reveals distinct roles for *Myh9* and *Myh10* in shaping the pial niche of the developing cortex (**Fig 8**).

## Discussion

Subcellular control is fundamental for functions of diverse cells and tissues, including those of the nervous system. A key question is how subcellular specificity is conveyed. By employing in vivo proximity labeling, we have discovered a local proteome of RGC basal endfeet, opening up a new understanding of this subcellular structure. To demonstrate the power of these data, we functionally interrogated MYH9 and MYH10. Surprisingly, these 2 NMHC II isoforms exhibit contrasting and complementary functions in modulating RGC complexity and endfoot position over the course of development. We further show that RGC morphology and adhesion to the BM non-cell autonomously directs interneuron number and position at the pia.

Taken together, our study illustrates how 2 protein-coding isoforms are differentially utilized spatially and temporally in RGCs to help instruct cortical architecture.

## In vivo proximity labeling uncovers first subcellular proteome of radial glia

Basal endfeet are evolutionarily conserved structures across progenitors, yet their composition and function has been largely enigmatic. Biological characterization of endfeet is technically challenging due to the inability to purify this tiny subcellular compartment. By combining in vivo proximity labeling of RGCs with mechanical dissection of the pia, we circumvented this limitation and produced an unbiased subcellular proteome of radial glia. We discovered that endfeet are enriched for components of ECM and ubiquitin-proteasome pathway, and cytoskeletal regulators. This screen thus nominates new candidates that may influence cortical development via localized control of RGCs.

Why are proteins subcellularly enriched in RGC basal endfeet? One answer lies in the complex, elongated morphology of RGCs that can extend several hundred micrometers in mice to centimeters in humans. This cellular architecture confers unique functions to apical and basal RGC structures. While the cell body controls gene expression and cell division, the basal processes and endfeet are critical for orchestrating migration of excitatory neurons from the VZ to the CP. Furthermore, as the cortex expands over the course of development, RGCs elongate their basal process and endfeet become more complex [4]. Thus, signaling and cytoskeletal components could be uniquely important for the acute remodeling and maintenance of these basal structures, as illustrated by our study of myosins. Endfoot-enriched proteins may be especially important in gyrencephalic species such as ferrets or in primates where RGC morphology is highly complex and the basal process extremely long [10,67]. Additionally, basal endfeet are conserved structures found in oRGs, a predominant neurogenic precursor of primates implicated in human brain development [6,7]. Thus, in future studies, it will be interesting and important to define the composition of basal endfoot proteomes of RGCs and oRGs across species.

Endfoot enrichment of proteins may be achieved by at least 2 main mechanisms. The mRNA encoding a specific protein may be trafficked to basal endfeet where it undergoes local translation, as we have previously shown can occur [27]. In support of this mechanism, we demonstrated mRNA localization for several proteins from our screen. This includes transcripts that were not previously identified [27], such as *Myh9* and *Myh10*. Further, a subset of this endfoot proteome is encoded by known FMRP-bound transcripts in endfeet [27]. The endfoot proteins that were not identified in this FMRP interactome may be encoded by transcripts bound by other RNA binding proteins. However, a second possible mechanism is that these proteins may be translated in the cell body and then trafficked to basal endfeet. We posit that both mechanisms are at play to control RGC basal structures.

## Extracellular matrix and ubiquitin components are enriched in basal endfeet

Using a cytoplasmic BirA*, we were surprised to discover ECM, including laminins, fibrillar collagens, and integrin interactors, as the largest category of enriched proteins in basal endfeet. Of note, ECM proteins are subcellularly represented at the mRNA and protein level in neuronal synapses [36,37]. Thus, across elongated cell types, local production of ECM components may be conserved [68]. We speculate that RGC basal endfeet may serve as hubs for local production and secretion of ECM components. Reciprocally, ECM may provide a physical and signaling niche to influence RGC endfeet, scaffold structure, or neurogenesis [13,69]. For example, germline knockout of *Col3a1*, which encodes an endfoot enriched protein, impairs

RGC morphology and causes neuronal heterotopias [70]. Future studies will be valuable for understanding how RGCs contribute to ECM production and function as well as the interplay within this niche. This question is relevant for human biology as ECM is instrumental for human cortical development. Indeed, ECM transcripts are especially elevated in developing human cortices compared to mice [71,72]. Furthermore, ECM is thought to help maintain the proliferative capacity of oRGs and influence cortical folding [7,73–75].

Ubiquitin and proteasome components including E3 ubiquitin-ligase RNF13 and proteasome subunits PSME1, PSME2, and PSMA7 were also enriched in endfeet. This suggests that beyond local translation, endfeet may be competent for ubiquitin-mediated degradation of subcellular proteins. We speculate local degradation could fine-tune protein levels including those that necessitate temporal restriction such as signaling factors. Beyond tagging proteins for degradation, ubiquitination can also promote protein trafficking, endocytosis, and cell division [76]. These additional roles may be important for endfoot maintenance and function.

## Non-muscle myosin isoforms control diverse aspects of RGC morphology

We demonstrate that local function of cytoskeletal regulators is essential for dynamic remodeling of RGC basal structures throughout cortical development. This is reflected by a focused analysis of *Myh9* and *Myh10* and also hinted at by the presence of additional cytoskeletal regulators in the local proteome.

*Myh9* was especially important for RGC endfoot morphology and position. NM II can negatively regulate cellular protrusion by modulating actin retrograde flow and pausing [77–79]. In fibroblasts, endothelial cells, and monocytes, NM IIA inhibition results in failure to retract processes [80,81]. Likewise, MYH9 may control basal endfoot protrusion via similar mechanisms. The mechanical force provided by NM II may also be used for formation and remodeling of ECM, including collagen and fibronectin [82–86]. Although we did not observe an overt defect in ECM in *Myh9* or *Myh10* mutants, altered ECM remodeling and/or production could contribute to aberrant protrusion of RGC basal endfeet through the BM.

In contrast to *Myh9*, we show that *Myh10* is necessary for maintaining attachment of both basal and apical endfeet. While *Myh10* cKO endfeet were initially associated with the BM, this attachment is progressively lost over development indicating an adhesion defect. Adhesion proteins, including tensin and talin, are enriched in endfeet, and integrin is required for the attachment of RGC endfeet to the BM [13,14,87]. Thus, MYH10 could influence endfoot attachment by modulating these proteins. In this regard, mechanical force generated by NM II can reverse steric inhibition of key adhesion components, including talin and integrin [88–90]. NM II can also cluster vinculins, talins, and integrins at the cell membrane by bundling actin filaments [91]. Additionally, NM II can promote nascent adhesion maturation and maintenance by attaching to adhesion-associated actin bundles [77,92,93]. Furthermore, NM II may influence endfoot attachment to the BM by remodeling ECM [82–86]. Taken together, this suggests several potential mechanisms by which MYH10 may promote endfoot adhesion.

Surprisingly, *Myh9* and *Myh10* conditional knockout mice had strikingly different RGC endfoot phenotypes. Our findings beg the question: How do these 2 protein isoforms mediate diverse roles in RGCs? One explanation is they are functionally interchangeable and that distinct phenotypes are explained by their subcellular localization rather than isoform-specific capabilities. In support of this possibility, in other contexts, some *Myh9*- or *Myh10*-specific phenotypes are explained by spatial or temporal expression differences rather than unique functions [94,95]. The smiFISH analysis of *Myh9* and *Myh10* across development revealed that the NMHC II isoforms have complementary temporal localization patterns. Thus, the unique phenotypes may be driven by developmental timing when each isoform is enriched in endfeet.

Perhaps NMHC II is required early in development to control RGC complexity and endfoot position, and later in development to regulate endfoot attachment to the BM. Therefore, due to their temporal localization patterns, MYH9 may mediate these earlier functions, whereas MYH10 mediates later functions. MYH9 and MYH10 share 77% sequence homology at the protein level, suggesting that they could, in fact, be interchangeable. Indeed, in the cell body of RGCs, we observe clear compensation by these isoforms. NMHC II A and B are each essential for cytokinesis; however, we did not observe proliferation defects in *Myh9* or *Myh10* cKO mice at E16.5 [96–98]. It remains possible that this compensation changes over the course of development and that neurogenesis defects may be present at later stages.

Alternatively, the differential functions of NMHC II proteins in endfeet could also be conveyed by isoform-specific properties. NMHC II isoforms do not heterodimerize and each has unique kinetic properties [42,99]. NM IIA generates quick actin contractions and can move filaments at least 3× faster than NM IIB [100–102]. NM IIB can bear higher mechanical loads than NMIIA, as it uses less energy to maintain long-term tension [101,103,104]. These kinetic properties may be advantageous for distinct regulation of branching and adhesion by *Myh9* and *Myh10*, respectively. Furthermore, MYH9 and MYH10 have differential phosphorylation sites in their coiled-coil and tail domains that may regulate their activity and ability to form filaments [105–109]. In sum, our data indicate that isoform switching, both temporally and subcellularly, can fine-tune molecular responses within endfeet.

## RGC morphology and endfoot position regulate cortical composition and organization

Our study provides evidence for new functions of RGC endfeet in cortical development. Previous genetic models of endfoot perturbation suggest their involvement in BM maintenance and prevention of neuronal over migration [11,12,14]. We did not observe such phenotypes in E16.5 *Myh9* and *Myh10* conditional knockout models, however. We hypothesize that this could be due to the timing of the endfoot detachment. Perhaps endfeet are necessary for ECM secretion to build the BM early in development and are no longer vital for this role at E15.5 and E16.5 when the majority of endfeet are detached in *Myh10* mutants. Alternatively, it is possible that 2 days of endfoot detachment (E14.5 to E16.5) is not sufficient to observe loss of the BM, which may be visible later in development. Here, we discover critical yet specific functions for endfeet in controlling proper number of interneurons in the MZ. Loss of *Myh10* led to 40% more interneurons in the MZ compared to Ctrl but did not affect transiently migratory CR cells. This makes sense, given that CR cells migrate very early into the MZ around E12.5 and are dynamic. Hence, a perturbation after that stage could be compensated for by reintroduction of new CR cells [65]. We further show that less endfoot complexity or absence of endfeet reduces interneurons in contact with BM. Both *Myh9* and *Myh10* mutants affected the fraction of LHX6+ cells in contact with the BM, consistent with other findings from our group [28].

How do endfeet control interneuron number and organization? Endfeet may provide both chemical and physical cues to influence interneurons. Previous studies have demonstrated that interneuron migration is regulated by extracellular signals from the meninges such as CXCL12 [110]. Endfeet could be an additional source of similar attractive or repulsive signals. Thus, in the absence of endfeet in *Myh10* cKOs, such cues may be missing resulting in excess interneuron number in the MZ. Additionally, as interneurons migrate tangentially through the MZ, they come in close contact with RGC basal structures [20,111]. Therefore, these transitory neurons may use endfeet to both establish proper position and number in the MZ. In *Myh10* cKOs, where endfeet are absent, we show that interneurons fill this "empty space." It is also

possible that the lack of a scaffold from RGC endfeet and basal processes prevents the radial migration of interneurons from the MZ into the cortex, thus retaining them in the MZ. Indeed, the cause and result of increased interneurons in the MZ should be explored in future studies. For example, it will be important to understand if there are rostral/caudal or medial/lateral differences in interneuron distribution, and whether the increase of interneurons in the MZ is accompanied by a corresponding decrease of interneurons in the CP. Overall, our data suggest that endfeet in the MZ are necessary to limit the number of cortical interneurons within this future synaptic zone.

In comparison, in *Myh9* cKOs, where RGC endfeet are present but disorganized, we posit that interneurons may not receive proper chemical or physical cues to attach to the BM. In future studies, it will be interesting to consider the longer-term consequences of interneuron positioning defects in both mutants. Mutations in NMHC II proteins are also linked to neurological disease, and, thus, it will be important to further understand the human relevance of these endfoot and interneuron defects [112,113].

Our study raises interesting directions for future investigation, particularly regarding roles of *Myh9* and *Myh10* in neurogenesis across development as well as the mechanisms underlying basal endfeet attachment. It is possible that loss of *Myh9* and/or *Myh10* does impact neurogenesis and neuronal positioning, either directly or indirectly as a result of basal RGC displacement (*Myh10*), but that this is only evident after E16.5. Regardless, we argue that neurogenesis roles would not explain the RGC endfeet and interneuron phenotypes, which present between E14.5 and E16.5. The *Myh10* cKO model will also enable future studies of the interplay between apical and basal endfeet to determine if the detachment of apical endfeet influences attachment of basal endfeet, and vice versa. If apical and basal endfeet attachments are independent of each other, it will be interesting to understand the unique and/or conserved mechanisms of endfoot adhesion.

This work furthers our understanding of local gene regulation in RGCs by discovering the first subcellular proteome of RGCs and uncovering endfoot-enriched proteins. Our study further argues for the importance of studying subcellular proteomes in understanding cell biology. While BirA* fusion proteins have been used for discovery of synaptic proteomes [114], we implement a new use of untagged BirA* to discover unbiased cytoplasmic proteomes in vivo. Hence, beyond cortical development, this approach for subcellular proteomics may be valuable in other cell types, such as neurons and glia. In sum, our study illustrates how discovery of subcellular proteomes can give unique insights into cellular and tissue-level processes.

## Materials and methods

### Mouse husbandry

All animal use was approved by the Duke Institutional Animal Care and Use Committee on 4/1/22 under approval number A060-22-03. The following mouse lines were used: *Myh9* $^{lox/lox}$ [47] (Gift, Robert Adelstein), *Myh10* $^{lox/lox}$ [51] (Gift, Robert Adelstein), *Emx1*-Cre (005628) [48] (Jackson Laboratory), *Nestin*-EGFP [46] (Gift, Qiang Lu), *Myh14*-GFP [40] (Gift, Terry Lechler). Proteomics and validation were performed on wild-type C57BL/6J mice (Jackson Laboratory). Background: *Myh9* $^{lox/lox}$ and *Myh10* $^{lox/lox}$ mixed C57BL/6J; SV129, *Emx1*-Cre *and Nestin*-EGFP C57BL/6J. Embryonic stage E0.5 defined as the morning the plug was identified. Control mice for *Myh9* and *Myh10* experiments were *Myh9* $^{lox/lox}$; *Emx1*- Cre$^{+/+}$ and *Myh10* $^{lox/lox}$; *Emx1*-Cre$^{+/+}$, respectively.

### Statistical methods and rigor

Description of specific n, statistical tests, and *p*-values for each experiment are reported in the figure legends. For all experiments, male and female mice were used and littermates were

compared when possible. All analyses were performed by 1 or more investigators blinded to genotype and/or condition.

## In utero biotinylation

The pcDNA3.1-MCS-BirA(R118G)-HA construct (gift, Scott Soderling) was subcloned into the pCAG-Ex2 vector. DNA for experiments was prepared using either the GenElute HP Endotoxin-Free Maxi Prep Kit (Sigma, NA0410) or the EndoFree Plasmid Maxi Kit (Qiagen, 12362). For IUEs, pregnant dams were kept under anesthesia with an isoflurane vaporizer. After making an incision in the abdomen, uterine horns were exposed. Using a micropipette, 1 μl of the plasmid solution was injected into the lateral ventricles of the embryonic brains. Subsequently, brains were electroporated using 5 pulses at 40 to 60 V for 50 ms at 950-ms intervals with platinum-plated BTX Tweezertrodes. The plasmid mixture consisting of pCAG-Ex2-BirA-HA at 1.0 μg/μL, pCAG-EGFP at 1.0 μg/μL, and FastGreen FCF Dye (Sigma, 861154) in PBS was injected into C57BL/6J embryos at E14.5. Embryos were dissected approximately 30 hours later at E15.5. Embryos were exposed to exogenous biotin by IP injection of the dam 3 times with 10 mM biotin/PBS (pH 7.5) at E13.4, E14.5 (at the time of IUE), and E15.5 (shortly before dissection). Electroporated brains were harvested in cold, sterile PBS. The electroporated regions were microdissected using a fluorescent microscope to visualize the GFP signal, and endfeet were isolated by mechanically separating the meninges from the cortex, as previously [27]. For the whole cortex sample, non-peeled brains containing the meninges and endfeet were used. For the no BirA* samples, non-electroporated littermate embryos were used. Tissues were stored at −80˚C until all samples had been collected.

## Affinity purification

Samples were lysed in 0.5 mL of homemade RIPA lysis buffer consisting of 50 mM Tris–HCl (pH 7.5), 150 mM NaCl, 1% sodium deoxycholate, 1% NP-40, 0.1% SDS, and 1 mM EDTA supplemented with Halt Protease Inhibitor Cocktail (Thermo Scientific, 78430) on ice for 10 minutes. They were then homogenized with an electric homogenizer on ice for two 15-second pulses separated by a 30-second rest. A total of 250 units of benzonase nuclease (Sigma Aldrich, E8263) were added and incubated on ice for 60 minutes to reduce lysate viscosity. Samples were sonicated on ice at 30% amplitude for 2 pulses of 15 seconds separated by a 60-second rest. The lysate was cleared by centrifugation at 21,000 rcf for 60 minutes at 4˚C. Cleared lysate was transferred to a tube with 50 μl NeutrAvidin Agarose beads (Thermo Scientific, 29200) in RIPA buffer. These tubes were incubated overnight on a rotator at 4˚C. After the overnight incubation, the flow-through was removed and beads proceeded with a series of 8-minute washes. The beads were washed twice with wash buffer 1 (2% SDS), once with wash buffer 2 (1% deoxycholate, 1% Triton X-100, 500 mM NaCl, 1 mM EDTA, and 50 mM HEPES (pH 7.5)), once with wash buffer 3 (250 mM LiCl, 0.5% NP-40, 0.5% deoxycholate, 1 mM EDTA, and 10 mM Tris (pH 8.1)), and twice with wash buffer 4 (50 mM Tris (pH 7.4) and 50 mM NaCl). Finally, the beads were boiled with 50 μL of Laemmli sample buffer, 50 mM DTT, and 40 mM biotin to elute the purified biotinylated proteins. The first quantitative mass spec analysis failed to detect many proteins because of deficiencies with the streptavidin beads. The purification was repeated with fresh beads and the flow-through from the first purification that had been stored at −80 before resubmitting for a successful second round of mass spec. The qualitative proteomics experiment was performed on 8 pooled cortices (whole cortex) from 2 electroporation litters for each condition (n = 1 technical replicate). For the quantitative proteomics experiment, endfoot preps were pooled from 66 to 68 cortices for each technical

replicate ($n$ = 3) and non-microdissected cortices from 6 embryos were pooled for each technical replicate ($n$ = 3). The samples were derived from a total of 51 electroporated litters.

## Western blot and silver stain

Six microdissected cortices were used for both BirA+ and BirA− conditions in the protein gel analysis. The affinity purified sample was split such that 60% (approximately 3.6 cortices) was used for silver stain analysis and 20% (approximately 1.2 cortices) for western blot analysis. Gradient 4% to 20% Mini-PROTEAN TGX polyacrylamide gels (Bio-Rad, 4568094) were used. Silver staining was performed with the Pierce Silver Stain Kit (Thermo Scientific, 24612) according to manufacturer's instructions. For western blotting, the electrophoresed gel was transferred to Trans-Blot TURBO PVDF membranes (Bio-Rad, 1704157), blocked in 1% BSA/TBST for 1 hour at room temperature, and incubated with HRP-conjugated streptavidin (Thermo Scientific, SNN1004, 1:10,000) in 1% BSA/TBST for 1 hour at room temperature. The blot was washed 3 times with TBST and developed with Pierce ECL Substrate (Thermo Scientific, 32106) in a Gel Doc XR+ (Bio-Rad, 1708195).

## Quantitative proteomics

For each sample, 50 μL was loaded onto an Invitrogen NuPAGE 4% to 12% SDSPAGE gel and run for approximately 5 minutes to electrophorese all proteins into the gel matrix. The entire molecular weight range was then excised in a single gel-band and subjected to standardized in-gel reduction, alkylation, and tryptic digestion. Following lyophilization of the extracted peptide mixtures, samples were resuspended in 12 μL of 2% acetonitrile/1% TFA supplemented with 12.5 fmol/μL yeast ADH. From each sample, 3 μL was removed to create a QC Pool sample that was run periodically throughout the acquisition period. Quantitative LC/MS/MS was performed on 4 μL of each sample, using a nanoAcquity UPLC system (Waters) coupled to a Thermo QExactive HF high-resolution accurate mass tandem mass spectrometer (Thermo) via a nanoelectrospray ionization source. Briefly, the sample was first trapped on a Symmetry C18 20 mm × 180 μm trapping column (5 μl/min at 99.9/0.1 v/v water/acetonitrile), after which the analytical separation was performed using a 1.8-μm Acquity HSS T3 C18 75 μm × 250 mm column (Waters) with a 90-minute linear gradient of 5% to 40% acetonitrile with 0.1% formic acid at a flow rate of 400 nanoliters/minute (nL/min) with a column temperature of 55˚C. Data collection on the QExactive HF mass spectrometer was performed in a data-dependent acquisition (DDA) mode of acquisition with a r = 120,000 (@ m/z 200) full MS scan from m/z 375 to 1,600 with a target AGC value of 3e6 ions followed by 12 MS/MS scans at r = 30,000 (@ m/z 200) at a target AGC value of 5e4 ions and 45 ms. A 20-second dynamic exclusion was employed to increase depth of coverage. The total analysis cycle time for each sample injection was approximately 2 hours.

## Proteomic bioinformatic analysis

Following 16 total UPLC-MS/MS analyses (excluding conditioning runs but including 4 replicate QC injections), data were imported into Rosetta Elucidator v 4.0 (Rosetta Biosoftware), and analyses were aligned based on the accurate mass and retention time of detected ions ("features") using PeakTeller algorithm in Elucidator. Relative peptide abundance was calculated based on area under the curve (AUC) of the selected ion chromatograms of the aligned features across all runs. The MS/MS data were searched against a custom Swissprot database with Mus musculus taxonomy (downloaded in April 2017) with additional proteins, including yeast ADH1, bovine serum albumin, and *E. coli* BirA, as well as an equal number of reversed-sequence "decoys" for false discovery rate determination. Mascot Distiller and Mascot Server

(v 2.5, Matrix Sciences) were utilized to produce fragment ion spectra and to perform the database searches. Database search parameters included fixed modification on Cys (carbamidomethyl) and variable modifications on Asn and Gln (deamidation). After individual peptide scoring using the PeptideProphet algorithm in Elucidator, the data were annotated at a 1.1% peptide false discovery rate. GO analysis was performed using the Database for Annotation, Visualization, and Integrated Discovery (DAVID) [115,116]. Molecular function (MF) direct results were reported for categories with $p \leq 0.05$. STRING analysis was performed using version 11.5. Network constructed with medium confidence option. Network edges represent confidence with line thickness equal to the strength of data support [35].

## Immunofluorescence

In utero BioID brains were fixed prior to microdissection for immunofluorescence analysis. Briefly, embryonic brains were fixed overnight in 4% formaldehyde/PBS, rinsed with PBS, incubated in 30% sucrose/PBS overnight, and then embedded in NEG-50 (Thermo Scientific, 6052). Unless otherwise specified, 20 μm sections were used for all staining. For antibody staining, sections were thawed, rinsed in PBS, permeabilized with 0.25% Triton X-100/PBS, blocked with 5% NGS/PBS, incubated with primary antibody in block buffer overnight at 4°C or room temperature for 2 hours, rinsed 3 times with PBS, incubated with species-specific secondary antibody (Alexa Fluor conjugated, Thermo 1:500) and DAPI in block buffer for 30 minutes or 2 hours at room temperature (LHX6 and p73), rinsed 3 times with PBS, and mounted with Vectashield Anti-Fade Mounting Medium (Vector Labs, H-1000). The following primary antibodies were used: <u>Rabbit</u>: anti-HA (Santa Cruz, sc-805, 1:250), anti-MYH9 (Biolegend, 909802 1:1,000), anti-MYH10 (Biolegend, 909902 1:1,000), anti-p73 (Cell Signaling, 14620S 1:250), anti-Calretinin (Swant, CR7697 1:1,000), anti-ISG15 (Thermo, 703132 1:100), anti-FERMT3 (Proteintech, 18131-1-AP 1:100), anti-TNS3 (Invitrogen, PA5-63112 1:75), anti-CC3 (Cell Signaling, 9661 1:400), anti-Ki67 (Cell Signaling, 12202 1:250), anti-Laminin (Millipore, AB2034 1:500), Anti-Collagen I (Invitrogen, PA5-95137 1:500), Anti-PSEM2 (ProteinTech, 12937-2-AP, 1:100) <u>Mouse</u>: anti-P150 Dynactin (BD Biosciences, 610473, 1:200), anti-LHX6 (Santa Cruz, sc-271 433, 1:500), anti-Reelin (Millipore, mab5364 1:100) <u>Rat</u>: anti-SOX2 (Thermo, 14-9811-82 1:500) <u>Other</u>: Streptavidin-Alexa Fluor 594 conjugate (Thermo Scientific, S11227, 1:500).

## Single-molecule inexpensive fluorescence in situ hybridization (smiFISH)

smiFISH was performed as described previously [117]. Briefly, 24 equimolar probes were hybridized to flap sequences conjugated to Cy3 (IDT). Brain sections were permeabilized with 0.5% triton in DEPC PBS for 30 minutes, washed twice with smiFISH wash buffer, and incubated with the hybridized probes overnight at 37 degrees. *Myh9*, *Myh10*, *Col4a1*, and *Myl6* probes were used at a dilution of 1:50 in smiFISH hybridization buffer. *Ccnd2* and *Psme1* probes were used 1:200. The following day, samples were washed twice in smiFISH wash buffer, incubated with 1:1,000 Hoechst (Thermo), washed with DEPC PBS, and mounted with vectashield (Vector Labs).

## CFSE labeling of RGC morphology

CFSE (cell trace, Thermo) was diluted to 20 μM in DMSO then mixed with PBS and Fast Green Dye to a final concentration of 2.5 μM. Pregnant mice were anesthetized as described above in the IUE section. To fill both ventricles, 2 μl CFSE solution was injected into the brains of embryonic mice (E13.5-E16.5). Embryos were returned to the abdomen and brains were collected 2 hours later.

### En face imaging of apical endfeet

We microdissected the cortices of E14.5 brains and fixed them overnight in 4% PFA. Stained floating in 24-well plates. Permeabilized 0.3% triton in PBS for 20 minutes, blocked in 5% NGS in PBS, primary antibody diluted in 5% NGS in PBS (β-catenin 1:200, ProteinTech, rhodamine-phalloidin 1:200, Molecular Probes) for 36 hours at 4°C. Secondary and Hoechst in 5% NGS in PBS (Alexa Fluor goat anti-rabbit 488 1:400, Thermo, Hoechst 1:1,000, Thermo) at 4°C overnight. Imaged with Andor Dragonfly Spinning Disk microscope with 63x oil immersion objective.

### Imaging and analysis of neurogenesis

*Myh10* en face apical endfoot images, *Myh9* 3D reconstructions, and *Myh9* LHX6/Laminin images were captured using an Andor Dragonfly Spinning Disk Confocal plus with 40X and 63X objectives. All other images were captured using a Zeiss Axio Observer Z.1 with Apotome for optical sectioning. 20X, 40X, and 63X objectives were used. For each experiment, 3 to 4 sections per embryo were imaged. Identical exposures, apotome phase images, and Z intervals were used. Cell counting was performed manually (FIJI cell counter) or automatically (QuPath). For QuPath quantification, the following parameters were used: requested pixel size = 0.1 μm, background radius = 5 μm, minimum area = 10 μm², maximum area = 200 μm², cell expansion = 2 μm, include cell nucleus and smooth boundaries were unselected. For quantification of cells touching the BM, cells colocalizing with the BM label (Laminin or Collagen) were counted manually. Cortical, SOX2, MZ, and Calretinin thickness measurements were taken using the line tool in ImageJ.

### Analysis of RGC basal process branching complexity and endfoot position

RGC branching complexity was quantified as previously described [28]. Briefly, Ctrl and cKO RGCs were sparsely labelled with pGLAST-EGFP-CAAX by IUE. Approximately 20 to 40 μm Z-stack images, 0.25 μm per step, were acquired on an Andor Dragonfly Spinning Disk Confocal microscope at 63X. 3D projections were then generated for assessment of endfoot number and position, and quantification of branches of each order per individual RGC in ImageJ. We count branches >5 μm. Round-shaped branch ends thicker than the processes were considered endfeet and categorized as attached, unattached, or protruded relative to the Laminin staining of the BM.

### Measurement of MZ thickness and cells in the MZ

The MZ was identified by the sparse nuclei between the pia and layer 2 for measurements of MZ thickness and quantification of cells within the MZ. For quantification of MZ thickness, 3 measurements were taken from each cortical section and averaged. Three sections were quantified per brain. For quantification of cells in the MZ (DAPI+, P73+, or LHX6+) cells within the MZ (identified by density) were counted by hand. For each image, the number of cells quantified was normalized by the length of the BM captured such that it was representative of 300 μm.

## Supporting information

**S1 Fig. Qualitative analysis of proteins labeled by unfused, cytoplasmic BirA* in vivo. (A)** Gradient 4% to 20% Silver stain gel of E15.5 whole RGC BioID− and BioID+ samples following affinity purification shows more biotinylated proteins in the BioID+ condition. Pool of 8 cortices per condition. Asterisks denote endogenously biotinylated carboxylases. (**B**) Cartoon

representation of the diverse categories of proteins labeled by cytoplasmic, untethered BirA*.
(TIF)

**S2 Fig. Validation of endfoot-enriched genes by immunofluorescence and smiFISH.** (**A**)
Cortical column showing MYH14 (green) expression at E15.5 marked by MYH14-EGFP
mouse. Dotted lines represent pial and ventricular borders. Asterisks denote background signal from the meninges. (**B**) Colocalization of MYH14-EGFP with mCherry-labeled endfeet
(red) shown by orthogonal views. (**C**) Anti-dynactin1 labeling throughout the cortex and at
the pia. Dotted box outlines region of interest for zoom (Right). (**D-G**) Protein (red) colocalization with EGFP-labeled endfeet (green) shown by orthogonal views. Endfeet outlined by
white dashed lines. ISG15 (**D**), FERMT3 (**E**), TNS3 (**F**), PSME2 (**G**). colocalization with
EGFP-labeled endfeet (green) shown by orthogonal views. (**H**) *Psme1* smiFISH (grey) colocalization with *Ccnd2* smiFISH (green) labeling endfeet shown by orthogonal views. Endfeet outlined by white dashed lines. (**I, J**) *mRNA* (grey) colocalization with EGFP labeled endfeet
(green) by orthogonal views. Endfeet outlined by white dashed lines. *Col4a1* (**I**) and *Myl6* (**J**).
*n* = 2 to 3 brains, 3 sections per brain (**A-J**). Scale bar: (**A**) 50 μm; (**B**) 10 μm; (**D-J**) 5 μm.
(TIF)

**S3 Fig. Characterization of *Myh9* cHet and cKO mice.** (**A**) MYH9 expression (red) in Ctrl
and *Myh9* cKO cortical columns at E16.5. Dotted lines represent pial and ventricular borders.
*n* = 3 per genotype from 2 litters. (**B**) *Myh9* expression by smiFISH (grey) in Ctrl and cKO
brains at E16.5. Yellow dashed line represents area of interest expanded (Right). *n* = 3 per
genotype from 2 litters. (**C**) CC3 (red) and DAPI (blue) staining of Ctrl, *Myh9* cHet, and *Myh9*
cKO cortical columns at E16.5. Dotted lines represent pial and ventricular borders. *n* = 3 per
genotype from 2 litters. (**D**) Representative images of Ctrl and *Myh9* cKO endfeet (labeled by
GLAST-EGFP-CAAX in green) relative to the BM (labeled by laminin in red). *n* = 5 Ctrl from
3 litters and 3 cKO from 2 litters. Scale bar: (**A**, **C, B-left**) 50 μm; (**B-right)** 25 μm; (**D**) 10 μm.
(TIF)

**S4 Fig. Characterization of *Myh10* cHet and cKO mice. (A**) MYH10 expression (green) in
Ctrl, *Myh10* cHet, and *Myh10* cKO cortical columns at E14.5. *n* = 3 per genotype from 1 litter.
(**B**) CC3 (red) staining of Ctrl, *Myh10* cHet, and *Myh10* cKO cortical columns at E16.5. *n* = 5
per genotype from 3 litters. (**C**) KI67 staining (red) of Ctrl and Myh10 cKO brains at E16.5.
*n* = 4 Ctrl, 3 cKO from 2 litters. (**D**) Comparison of Ctrl and cHet en face apical endfeet labeled
with β-catenin (green) and Phalloidin (red). Note: Ctrl images are the same as those from Fig
5L. *n* = 9 Ctrl from 3 litters and 3 cHet from 2 litters. Dotted lines represent pial and ventricular borders. (**A**, **B**, **C**) Three sections imaged per brain (**A**, **B**, **C**). Scale bar: (**A**, **B**, **C**) 50 μm;
(**D**) 25 μm.
(TIF)

**S5 Fig. Loss of Myh10 from RGCs does not impact CR cells at E16.5 or MZ architecture at
E13.5.** (**A**) Reelin expression (green) in the MZ of Ctrl and *Myh10* cKO brains at E16.5. *n* = 6
per genotype from 2 litters. (**B**) Reelin expression (green) in the MZ of Ctrl and *Myh10* cKO
brains at E13.5. *n* = 3 per genotype from 1 litter. (**C**) Calretinin expression (red) in the MZ of
Ctrl and *Myh10* cKO brains at E13.5. *n* = 3 per genotype from 1 litter. Scale bar: (**A**, **B**, **C**)
50 μm. Dotted lines represent border between pia and MZ (**A**, **B**, **C**).
(TIF)

**S1 Table. Quantitative proteomics raw peptide data.**
(XLSX)

**S2 Table. Quantitative proteomics raw expression data.**
(XLSX)

**S3 Table. Quantitative proteomics intensity scaled data.**
(XLSX)

**S4 Table. Proteins enriched ≥1.5-fold in endfoot BioID+ vs. endfoot BioID−.**
(XLSX)

**S5 Table. Proteins enriched ≥1.5-fold in whole RGC BioID+ vs. Whole RGC BioID−.**
(XLSX)

**S6 Table. Proteins enriched ≥1.5-fold in endfoot BioID+ vs. endfoot BioID− with a *p*-value ≤ 0.28.**
(XLSX)

**S7 Table. Proteins enriched ≥1.5-fold in whole RGC BioID+ vs. whole RGC BioID− with a *p*-value ≤ 0.28.**
(XLSX)

**S8 Table. Proteins enriched ≥1.5-fold in endfoot BioID+ vs. endfoot BioID− with a *p*-value ≤ 0.28 and enriched ≥1.25-fold compared to whole RGC BioID+.**
(XLSX)

**S1 Data. Results and numerical values underlying summary data in Fig 2 graphs including panels Fig 2B–2F.**
(XLSX)

**S2 Data. Numerical values underlying summary data in Fig 4 graphs including panels Fig 4D–4F and 4I–4L.**
(XLSX)

**S3 Data. Numerical values underlying summary data in Fig 5 graphs including panels Fig 5G and 5H.**
(XLSX)

**S4 Data. Numerical values underlying summary data in Fig 6 graphs including panels Fig 6C, 6D, 6F and 6H.**
(XLSX)

**S5 Data. Numerical values underlying summary data in Fig 7 graphs including panels Fig 7B, 7D, 7F and 7H.**
(XLSX)

**S1 Raw Images. Uncropped versions of gels and blots in Figs 1F and S1A.**
(PDF)

## Acknowledgments

The authors thank past and present members of the Silver lab for discussions and careful reading of this manuscript. We also thank the Duke Proteomics and Microscopy core facilities. We thank Robert Adelstein, Terry Lechler, Qiang Lu, and Scott Soderling for sharing mice and reagents.

## Author Contributions

**Conceptualization:** Brooke R. D'Arcy, Ashley L. Lennox, Debra L. Silver.

**Data curation:** Brooke R. D'Arcy, Ashley L. Lennox, Debra L. Silver.

**Formal analysis:** Brooke R. D'Arcy, Camila Manso Musso, Annalise Bracher, Debra L. Silver.

**Funding acquisition:** Brooke R. D'Arcy, Ashley L. Lennox, Debra L. Silver.

**Investigation:** Brooke R. D'Arcy, Ashley L. Lennox, Camila Manso Musso, Annalise Bracher, Carla Escobar-Tomlienovich, Stephany Perez-Sanchez.

**Methodology:** Brooke R. D'Arcy, Ashley L. Lennox, Camila Manso Musso, Debra L. Silver.

**Project administration:** Debra L. Silver.

**Resources:** Debra L. Silver.

**Supervision:** Debra L. Silver.

**Validation:** Brooke R. D'Arcy, Camila Manso Musso, Annalise Bracher, Carla Escobar-Tomlienovich, Stephany Perez-Sanchez.

**Visualization:** Brooke R. D'Arcy, Ashley L. Lennox, Camila Manso Musso.

**Writing – original draft:** Brooke R. D'Arcy, Ashley L. Lennox, Debra L. Silver.

**Writing – review & editing:** Brooke R. D'Arcy, Ashley L. Lennox, Camila Manso Musso, Annalise Bracher, Carla Escobar-Tomlienovich, Stephany Perez-Sanchez, Debra L. Silver.

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
