## [Editor Report · Decision Letter 0]

14 Nov 2022

Dear Debbie, 

I was glad to see Brooke D'Arcy's manuscript entitled "Subcellular proteome of radial glia reveals non-muscle myosins control basal endfeet to mediate interneuron organization" land on my desk today for consideration as a Research Article by PLOS Biology.

We would like to send the work out for external peer review. However, before we can do so, we need you to complete your submission by providing the metadata that is required for full assessment. To this end, please login to Editorial Manager where you will find the paper in the 'Submissions Needing Revisions' folder on your homepage. Please click 'Revise Submission' from the Action Links and complete all additional questions in the submission questionnaire.

Once your full submission is complete, your paper will undergo a series of checks in preparation for peer review. After your manuscript has passed the checks it will be sent out for review. To provide the metadata for your submission, please Login to Editorial Manager (https://www.editorialmanager.com/pbiology) within two working days, i.e. by Nov 16 2022 11:59PM.

Kind regards,

Kris

Kris Dickson, Ph.D., (she/her)

Neurosciences Senior Editor/Section Manager

PLOS Biology

kdickson@plos.org

---

## [Decision Letter · Decision Letter 1]

4 Jan 2023

Dear Dr Silver,

Thank you for your patience while your manuscript "Subcellular proteome of radial glia reveals non-muscle myosins control basal endfeet to mediate interneuron organization" was peer-reviewed at PLOS Biology. It has now been evaluated by the PLOS Biology editors, an Academic Editor with relevant expertise, and by several independent reviewers. 

Based on the reviews and our editorial assessment, I am happy to say that we are likely to accept this manuscript for publication. As you will see, the reviewers were overall quite positive about the manuscript. We and our Academic Editor also tend to agree with Reviewer 3 that additional experiments would not change the overall thesis of this work. With this in mind, we ask that you address the various reviewer comments with textual revisions and the inclusion of a "limitations" paragraph that discusses potential future experiments that would help to address some of the comments that were raised (e.g. In particular - Reviewer 1 request to examine Myh9 loss in RGCs on neurogenesis at later stages and on the apical endfoot).

When revising your submission, we also ask that you consider a slight title change. While we appreciate that the study starts out with an unbiased screen, we feel the title would be stronger without this inclusion. We'd suggest instead simply going with: "Non-muscle myosins control radial glial basal endfeet to mediate interneuron organization"

Please also update your abstract to include reference to the model organism being used. Please also make sure to carefully address the other data and other policy-related requests listed at the bottom of this email.

As you address these items, please take this chance to review your reference list to ensure that it is complete and correct. If you have cited papers that have been retracted, please include the rationale for doing so in the manuscript text, or remove these references and replace them with relevant current references. Any changes to the reference list should be mentioned in the cover letter that accompanies your revised manuscript.

We expect to receive your revised manuscript within two weeks. 

*Published Peer Review History*

*Press*

Sincerely,

Kris

Kris Dickson, Ph.D., (she/her)

Neurosciences Senior Editor/Section Manager,

kdickson@plos.org,

PLOS Biology

DATA POLICY:

Thank you for providing the underlying data for your graphical figures, complying with the PLOS Data Policy, which requires that all data be made available without restriction: http://journals.plos.org/plosbiology/s/data-availability. 

For more information, please also see this editorial: http://dx.doi.org/10.1371/journal.pbio.1001797. Note that we do not require all raw data. Rather, we ask that all individual quantitative observations that underlie the data summarized in the figures and results of your paper be made available.

1) Additionally, please also ensure that figure legends in your manuscript include information on where the underlying data can be found, and ensure your supplemental data file/s has a legend.

SPECIES INDICATED IN THE ABSTRACT

- Please note that per journal policy, the model system/species studied should be clearly stated in the abstract of your manuscript. 

We require the original, uncropped and minimally adjusted images supporting all blot and gel results reported in an article's figures or Supporting Information files. We will require these files before a manuscript can be accepted so please prepare and upload them now. Please carefully read our guidelines for how to prepare and upload this data: https://journals.plos.org/plosbiology/s/figures#loc-blot-and-gel-reporting-requirements

1) The only thing that we were unable to locate was a full image of the Figure 1F western blot. Please provide this as a supplemental file when submitting your revision.

DATA NOT SHOWN?

- Please note that per journal policy, we do not allow the mention of "data not shown", "personal communication", "manuscript in preparation" or other references to data that is not publicly available or contained within this manuscript. Please carefully check your study for any such references and either remove mention of these data or provide figures presenting the results and the data underlying the figure(s).

Reviewer remarks:

Reviewer #1: No

Reviewer #2: No

Reviewer #3: No

Reviewer #1: In this study, D'Arcy et al. set out to define the local proteome of radial glial cell (RGC) endfeet in the developing mouse neocortex. To this end, they used a creative approach combining RGC endfoot microdissection, a method which allows for mechanical separation of the basal endfoot from the rest of the RGC, with BioID proximity-labeling proteomics. In doing so, they identified a cohort of proteins, including components of extracellular matrix, ubiquitin-proteasome pathway, and cytoskeletal regulators, that are enriched in RGC basal endfeet. Among them, the non-muscle myosin heavy chain II (NMHC II) isoforms, namely NMHC IIA (MYH9) and NMHC IIB (MYH10), were among the most abundant. This prompted the authors to further characterize the function of these proteins in RGC endfeet, taking advantage of Myh9lox/lox;Emx1-Cre and Myh10lox/lox;Emx1-Cre mutant mice . By performing loss of function studies, they found that Myh9 is important for the regulation of RGC complexity and endfoot organization relative to the basement membrane (BM) while Myh10 is required for maintaining RGC basal and apical endfeet attachments. Finally, the authors present data supporting that Myh9 non-cell autonomously regulates interneuron position in the marginal zone (MZ) while Myh10 regulates both interneuron position and number.

This is an interesting study, providing novel insight into the subcellular proteome of RGC basal endfeet. Furthermore, the study sheds novel light on the functions of Myh9 and Myh10 in the regulation of RGC endfoot morphology. The manuscript is well written, and overall the experiments are well executed. There are, however, a number of issues that need to be addressed (see below). 

Specific points:

1) By performing qualitative tandem mass spectrometry on proteins affinity purified from BioID+ and BioID- cortices, the authors found that besides BioID+ specific biotinylated proteins, there were 110 BioID- specific biotinylated proteins (almost 25% of the total biotinylated proteins found in BioID- samples) (Fig S1). Since the BioID- samples are representative of the baseline biotinylation in control mice, one would expect all biotinylated proteins to also appear in the BioID+ samples ? What is the identity of these proteins? The authors should comment on this. 

2) The authors show that loss of Myh9 in RGCs leads to decreased branching complexity and protrusion of basal endfeet through the BM. To assess that these defects are not secondary to cell body defects, they examined the effects of Myh9 loss on progenitor cell organization, apoptosis, and neurogenesis at E16.5. While their data support that the basal endfoot phenotypes are not likely the result of defects in these processes, they do not per se exclude a role for Myh9 in any of these processes, as they may be only apparent at later stages of development. The authors should at least include data examining the impact of Myh9 loss in RGCs on neurogenesis at later stages of development. Also, it is not clear whether loss of Myh9 has no effect on the apical endfoot. Magnified images of the apical endfoot and cell body of Myh9 cKO RGCs should be included. 

3) The authors present data showing that the basal endfeet of Myh10 cKO RGCs are initially attached at the pia (at E14.5) and then progressively become unattached from the BM (by E15.5). They also found that the apical endfeet of Myh10 cKO RGCs become disorganized and detached from the ventricle (starting at E14.5), leading to more basally located RGCs. Can the authors exclude that the gradual detachment of the basal endfeet is the result of a detachment of the apical endfeet? They should comment on this. Also, previous studies have shown that disorganization/detachment of RGC apical endfeet results in more basally dividing progenitors, ultimately leading to an increase in neurogenesis. The authors report that loss of Myh10 in RGCs does not affect neurogenesis at E16.5. Do they see altered neurogenesis at later developmental time points in the Myh10 cKO animals? If not, how do they explain this? Also, more representative images should be included for Fig 5J. Why is there hardly any staining seen for MYH10 at E15.5 while there is at E16.5?

4) The authors report that Myh9 and Myh10 transcripts localize to the basal endfeet with distinct temporal dynamics, with Myh9 mRNA being most enriched in the basal endfeet at E12.5 and declining at E13.5-E14.5 and Myh10 mRNA being low at early developmental stages (E12,5) and gradually increasing until the last stage examined (E16.5). Do the authors see similar results at the protein level? Of note, the authors should include more representative images for Fig. 3F, as the gradual increase of Myh10 mRNA over development is not very evident from the images shown. Also, the authors postulate that the Myh9 and Myh10 phenotypes may be driven by developmental timing when each isoform is enriched in the endfeet. They should further elaborate in the discussion how they envision the developmental timing could explain the divergent phenotypes observed for Myh9 and Myh10 cKO RGCs.

5) The authors present data supporting that Myh9 and Myh10 loss of function phenotypes are associated with altered distribution of interneurons in the marginal zone (MZ) and also increased interneuron number in the case of Myh10 loss. More representative images though should be included for Fig. 7G, and the authors should indicate that here the BM was labeled using anti-collagen I antibody. Perhaps beyond the scope of this study, it would be interesting to know the functional consequences of these alterations for cortical development and function. 

Reviewer #2: Radial glial cells perform many essential functions in the developing central nervous system. They are progenitors for neurons and glia, provide scaffolds for cell migration, give structural integrity to developing CNS structures, and help establish and maintain extracellular matrix structures like the basement membrane. Key to these functions of radial glial cells is their morphology. Radial glia form a bipolar pseudostratified epithelium, with their apical endfeet forming junctions at the ventricles and their basal endfeet forming attachments at the basement membrane between the brain and overlying meninges. These endfeet are important for several aspects of radial glia function, including their ability to attach to and remodel the basement membrane, maintain radial bipolar morphologies, and receive extracellular cues. However, most of the molecular mechanisms that establish radial glia endfeet and allow them to perform these functions are not known. An important barrier to progress in this area has been the small size of the endfeet compared to the rest of the developing brain, which limits access to study the molecules that are important for endfeet formation and function. In this manuscript, the authors develop a new approach for identifying proteins enriched in radial glia endfeet and test the functions of two such proteins in brain development.

Using biochemical, molecular, and genetic approaches in vivo, the authors make the following major, novel scientific findings: 1) Using in vivo proximity labeling in a novel way, they identify many proteins that are enriched in radial glia endfeet. 2) They validate several of these proteins by immunohistochemistry and some of their mRNAs by smiFISH, including non-muscle myosins MYH9 and MYH10 3) They show that conditional knockout of Myh9 in radial glia (and their progeny) causes defects in basal endfoot complexity and organization. 4) They find that conditional knockout of Myh10 causes a complete detachment of basal and apical endfeet. 5) They show that the endfeet defects caused by conditional knockout of Myh9 or Myh10 in radial glia cause non-cell-autonomous perturbations in interneuron organization. The authors conclude that these non-muscle myosins have distinct and complementary functions in radial glia endfeet organization and maintenance, and that proper endfeet function is required for normal organization of interneurons. 

This study is relevant to readers of PLOS Biology who are interested in brain development, non-progenitor roles of radial glia, control and function of cell polarity and morphology, and cell migration and adhesion. The study is also of broader relevance and interest to those who study mechanisms of RNA/protein trafficking in highly polarized cells, as well as those interested in using in vivo approaches to uncovering the proteomes of subcellular structures.

Overall, the experiments are well designed and executed. The data support the claims made, the presentation of the results is convincing, and the figures are easy to understand. The manuscript is well written and existing literature is appropriately cited. The methods are generally well-described. The study is outstanding for its creative use of in vivo biochemical approaches, state-of-the-art in vivo labeling and imaging of cells and subcellular structures, and solid genetic approaches.

In this reviewer's opinion, the following suggestions would improve the manuscript:

1. The "Control" genotype for genetic experiments is never defined in the results or methods. Should clarify the genotype(s) - fl/fl no cre, +/+ with cre, etc. 

2. Fig 6B-D - It is unclear how the MZ is defined. MZ thickness is typically assessed by the sparse nuclei situated between the pia and the densely populated top of layer 2. The image in 6B is cropped too much to tell if layer 2 is still obvious (more dense than the MZ) in the mutants? This should be clarified in the methods, along with how many measurements were taken (and presumably averaged) per brain. 

3. Fig 6C, H and Fig 7B, D - It should be clarified in the methods how cells in MZ were quantified. Presumably they were quantified per unit area, which should be reported (along with actual values, as in comment 4b below).

4. Quantifications and graphs: 

 a. Fig 4I and K - individual data points should be shown. In this case, a SuperPlot (see DOI 10.1083/jcb.202001064) would be more appropriate, in which the means for each brain (perhaps color-coded by litter) are plotted on top of the individual cells (color coded by brain). This will provide more information about possible brain to brain or litter to litter variability, beyond the cell to cell variability already reported in the current graph. 

 b. Fig 6H, 7B, and 7D - instead of presenting the data as "normalized by litter", consider presenting the actual values for each brain (n) in a SuperPlot. The data points for each brain can then be color coded (or otherwise represented, e.g., by different shapes) by litter, with averages of each litter superimposed (and possibly lines connecting littermates controls). 

5. The authors find more interneurons in the MZ of Myh10 mutants compared to Controls and conclude that more interneurons "flood into the open space of the MZ", possibly due to missing cues in the absence of radial glia endfeet. Although this is plausible, it would not be my first interpretation. Another possibility is that the INs that are migrating along their usual path in the MZ cannot subsequently migrate into the cortical plate properly. This could be because the radial glia scaffold and/or the general architecture of the cortical plate is perturbed. One simple way to assess this possibility would be to quantify the number of INs in the cortical plate and compare mutants to controls. If there are fewer INs in the cortical plate, this could mean that INs are getting stuck in the MZ (rather than "flooding" the MZ). If the number of INs in the cortical plate are not affected, it would be interesting to know where all the extra INs in the MZ came from. Is there a difference in their rostral-caudal or medial-lateral distribution? It would be informative and relatively straightforward to test these possibilities. At the very least they should be discussed as possible interpretations of the results. 

6. Althought not the main point of the manuscript, the authors speculate that RGC basal endfeet may serve as hubs for local production and secretion of ECM components (lines 523-524). In this regard, it would be valuable to the readers if the authors discuss the fact that the basement membrane (staining for Laminin and COL1 in Fig 7) is not perturbed in the Myh9 and Myh10 knockouts. This is especially surprising for the Myh10 mutants that don't have endfeet at all. Perhaps the ECM production/secretion functions happen prior to the ages at which the endfeet detach?

Minor points:

1. Fig 3 - In the legend it is unclear why "n=" is reported since there are no quantifications. Does this mean that the images shown are representative of that many brains? 

2. Most instances of the word 'which' throughout the document should be changed to 'that'. Examples: lines 18, 89, 103, 108, etc (but not 99, 126, 153, etc). 

3. Line 153 - should clarify "endfoot proteins by IF and <mRNAs> by smiFISH"

4. Lines 591-592 - An alternate interpretation is that the signal coordinating CR cell migration (CXCL12) is coming from the meninges, not the endfeet.

 a. Also relates to lines 597-598 when discussing interneuron migration. Since CR and IN migration are both regulated by CXCL12 from the pia (not endfeet), and CR position is unaffected, it seems unlikely that this signal is relevant to the Myh9 and Myh10 knockout phenotypes. 

Reviewer #3: This exciting manuscript convincingly demonstrates the first proteomic assessment of the end feet of RGCs and that MYH9 and MYH10 myosins have diverging functions in both endfeet branching complexity and end feet adhesion to the pial basement membrane.

While it is not surprising that MYH9 and MYH10 have diverging functions in this and other systems, the high quality of the work and the relevance neurological disorders where cellular interactions with neural cells with either the pial basement membrane and apical contacts in the VZ are common, and potential generalizability to other brain areas render this an outstanding contribution to our understanding of neural development.

With so many complex experiments in play, there are a fair number of issues that could be nit-picked, but this reviewer will refrain from doing so. This reviewer's opinion is that such suggestions would have minimal impact on the quality of the paper's message and would waste resources better left to the investigators for their next study. It would be interesting to compare and contrast adhesive mechanisms to the pial basement membrane and at apical attachments in the discussion section to more deeply explain some of the interesting phenotypes uncovered in the study.

---

## [Editor Report · Decision Letter 2]

17 Jan 2023

Dear Dr Silver,

Thank you for the submission of your revised Research Article "Non-muscle myosins control radial glial basal endfeet to mediate interneuron organization" for publication in PLOS Biology. On behalf of my colleagues and the Academic Editor, Cody Smith, I am pleased to say that we can in principle accept your manuscript for publication, provided you address any remaining formatting and reporting issues. These will be detailed in an email you should receive within 2-3 business days from our colleagues in the journal operations team; no action is required from you until then. Please note that we will not be able to formally accept your manuscript and schedule it for publication until you have completed any requested changes.

PRESS

Sincerely, 

Kris Dickson, Ph.D., (she/her),

Neurosciences Senior Editor/Section Manager

PLOS Biology

kdickson@plos.org